# GNEISSWEB: PREPARING HIGH QUALITY DATA FOR LLMS AT SCALE

**Hajar Emami Gohari**,* **Swanand Ravindra Kadhe**∗, **Yousaf Shah**, **Constantin M Adam**,
**Abdulhamid Adebayo**, **Praneet Adusumilli**, **Farhan Ahmed**, **Nathalie Baracaldo**,
**Santosh Subhashrao Borse**, **Yuan-Chi Chang**, **Xuan-Hong Dang**, **Nirmit Desai**, **Revital Eres**,
**Ran Iwamoto**, **Alexei A. Karve**, **Yan Koyfman**, **Wei-Han Lee**, **Changchang Liu**,
**Boris Lublinsky**, **Takuya Ohko**, **Pablo Pesce**, **Maroun Touma**, **Shiqiang Wang**,
**Shalisha Witherspoon**, **Herbert Woisetschläger**, **David Wood**, **Kun-Lung Wu**, **Issei Yoshida**,
**Syed Zawad**, **Petros Zerfos**, **Yi Zhou**, **Bishwaranjan Bhattacharjee**
IBM Research

## ABSTRACT

Data quantity and quality play a vital role in determining the performance of Large Language Models (LLMs). High-quality data, in particular, can significantly boost the LLM's ability to generalize on a wide range of downstream tasks. In this paper, we introduce **GneissWeb**, a large dataset of around 10 trillion tokens that caters to the data quality and quantity requirements of training LLMs. Our GneissWeb recipe that produced the dataset consists of sharded exact sub-string deduplication and a judiciously constructed ensemble of quality filters. GneissWeb goes beyond simple model-based quality filtering used in recent datasets by designing an ensemble of filters incorporating novel quality filters. Novel components enable us to achieve a favorable trade-off between data quality and quantity, producing models that outperform models trained on state-of-the-art open large datasets (5+ trillion tokens). We show that models trained using GneissWeb outperform those trained on FineWeb-V1.1.0 by 2.73 percentage points in terms of average scores on a set of 11 commonly used benchmarks (both zero-shot and few-shot) for pre-training dataset evaluation. When the evaluation set is extended to 20 benchmarks (both zero-shot and few-shot), models trained using GneissWeb still achieve a 1.75 percentage points gain over those trained on FineWeb-V1.1.0.

## 1 INTRODUCTION

Large Language Models (LLM) are becoming pervasive in many aspects of life. While it is widely accepted that the quality and quantity of training data play a critical role in dictating the performance of LLMs, the pre-training datasets for leading LLMs, such as Llama-3 (Grattafiori et al., 2024) and Mixtral (Jiang et al., 2024), remain inaccessible to the public at the time of writing of this paper. Opacity of datasets used to train leading LLMs has motivated the development of several open-source datasets (Penedo et al., 2023; Soboleva et al., 2023; Weber et al., 2024; Soldaini et al., 2024; Li et al., 2024). These datasets are mainly derived by processing text from the Common Crawl Crawl (2007) and optionally mixing some high-quality data sources (e.g., GitHub).

However, a majority of these datasets are less than 5 trillion (5T) tokens which limits their suitability for pre-training massive LLMs. Indeed, recent state-of-the-art LLMs have been trained on far more data than what the *Chinchilla* scaling laws (Hoffmann et al., 2022) would deem as optimal. For instance, Llama-3 (Grattafiori et al., 2024) family of models are trained on 15T tokens (compared to 1.8T tokens for Llama-2 (Touvron et al., 2023)), Gemma-2 (Team et al., 2024) family of models are trained on 13T tokens, and Granite-3.0 (Granite, 2024) family of models are trained on 12T tokens.

Large models typically undergo long token horizon pre-training consisting of two stages (Granite, 2024). In Stage-1 of pre-training, the model is trained on a very large corpus of data to cover the breadth, followed by a Stage-2 pre-training which uses much higher quality but comparatively

---

*Equal contribution.

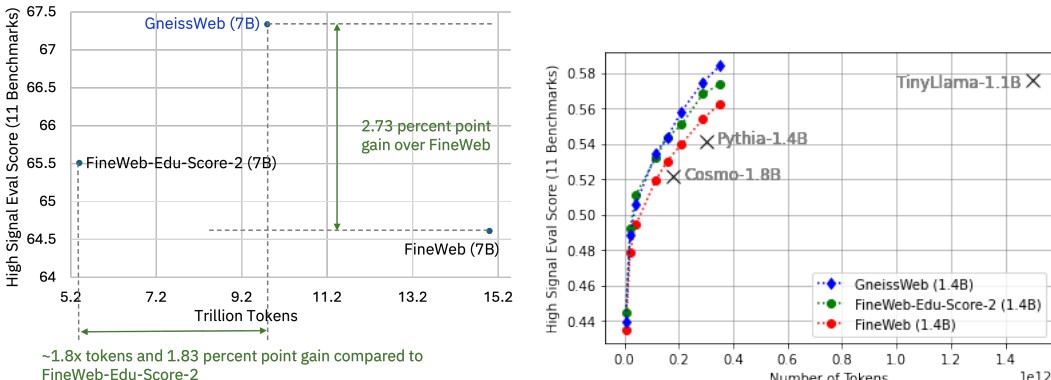

Figure 1: **GneissWeb (∼10T tokens) outperforms state-of-the-art open-source datasets with 5T+ tokens.** We compare average scores on a set of 11 benchmarks with 18 variants (zero-shot and few-shot) for 7B parameter models (left) and 1.4B parameter models (right) trained on 350B tokens, sampled randomly from each dataset. We also compare with state-of-the-art existing models of roughly 1B parameter size. Models trained on GneissWeb achieve higher performance than the models trained on other datasets and existing models. Detailed evaluations are in Section 4.

smaller dataset to improve performance. Their massive size demands make it challenging to develop high-quality pre-training datasets that are suitable for Stage-1 long token horizon training.

In this paper, we tackle a fundamental challenge in LLM training: constructing Stage-1 datasets that balance scale and quality for long-horizon pre-training. We introduce **GneissWeb**[1] dataset of around 10T tokens to fill a critical gap between small datasets (less than 5T tokens) and the largest ones (FineWeb at ∼15T (Penedo et al., 2024) and RedPajamaV2 at ∼30T tokens (Weber et al., 2024)). Our core contribution is GneissWeb recipe, a scalable, reusable Stage-1 dataset construction recipe which is built to match the token quantity and quality needs of Stage-1 pre-training by developing novel processing steps and quality filters that can effectively filter out low-quality data. The recipe consists of sharded exact substring deduplication and a judiciously constructed ensemble of quality filters.

Our work introduces several new components beyond existing approaches:

- We go beyond simple model-based quality filtering used in recent datasets and design an *ensemble of filters* incorporating *novel quality filters* based on characteristics of the text contents. Ensemble of quality filters enables us to achieve a fine-grained trade-off between the quality and quantity of the tokens retained.
- Our *novel readability score quality filter* effectively utilizes information based on human ability of reading documents from different domains for identifying and excluding low-quality documents.
- We develop a novel quality filtering called *Extreme-Tokenized Documents Removal* that effectively leverages information from both the "pre-tokenization" stage and the "post-tokenization" stage to filter out low-quality documents based on tokenized data.
- We leverage the domain information as *category* of a document in our quality filtering process which reduces the risk of losing high-quality data by processing all documents in the same way.
- These novel quality filters and the methodology of leveraging domain information can *also be used outside of the GneissWeb recipe* to enhance other data curation recipes.

Our evaluations demonstrate that GneissWeb outperforms state-of-the-art large open datasets of 5T+ tokens (see Figure 1). Specifically, 7B parameter models trained on GneissWeb outperform those trained on FineWeb-V1.1.0 of 15T tokens (Penedo et al., 2024) by 2.73 percent points in terms of average score computed on a set of 11 commonly used benchmarks (both zero-shot and few-shot), and by 1.75 percent points on an extended set of 20 benchmarks (see Section 4 for more details). GneissWeb performance is also superior at 1.4B and 3B model sizes compared to models trained on other large open datasets.

---

[1]*Gneiss*, pronounced "nice", is a durable igneous rock.

GneissWeb is fully prepared using an open sourced Data Prep Kit[2] (Wood et al., 2024), with the majority of data preparation steps efficiently running at scale on Kubernetes clusters. The entire GneissWeb recipe along with ablation models are publicly released[3].

**Related Work:** Over the past few years, the community has curated a number of pre-training datasets including Weber et al. (2024); Penedo et al. (2023; 2024); Soldaini et al. (2024); Li et al. (2024); Tang et al. (2024); Su et al. (2024); Tokpanov et al. (2024) (see Appendix A for details). A majority of the datasets are smaller than 5T tokens, limiting their suitability for long token horizon Stage-1 pre-training. A couple of exceptions are FineWeb (Penedo et al., 2024) (15T toknes) and RedPajama v2 (30T tokens) (Weber et al., 2024). FineWeb has been shown to outperform several prior public datasets including RedPajama v2 (see Appendix B for details).

Two smaller versions of FineWeb – FineWeb-Edu (1.3T tokens) and FineWeb-Edu-Score-2 (5.4T tokens) (Penedo et al., 2024), and the recent DCLM-Baseline (3.8T tokens) (Li et al., 2024) improve data quality over FineWeb, but they do so by performing aggressive model-based quality filtering. Such an aggressive filtering cuts down their size. These small data sizes are typically not sufficient for pre-training (as pre-training typically consists of only one pass or few passes over the pre-training dataset (Muennighoff et al., 2023)). In contrast, our GneissWeb recipe is designed to achieve a favorable trade-off between data quality and quantity, thereby producing ∼10T high quality tokens with higher performance than prior datasets with 5T+ tokens. Motivated by its sufficiently large quantity and high quality, we take FineWeb as the starting point to build our dataset.

## 2 THE GNEISSWEB RECIPE

We describe the *GneissWeb recipe* designed to distill ∼10T tokens high quality tokens from FineWeb. Even though we build on top of FineWeb, the recipe can be applied on top of other datasets and is not tied to FineWeb. In the following, we present the recipe ingredients along with key ablation experiments (with more details in Appendix C).

### 2.1 ABLATION AND EVALUATION SETUP

We train data ablation models that are identical in terms of architecture and training parameters, except for the data they were trained on.

**Training:** Following prior ablations in open datasets from Penedo et al. (2023; 2024); Li et al. (2024), we train decoder-only models with Llama architecture (Touvron et al., 2023). We typically train ablation models on 35B (slightly larger than the Chinchilla optimal) tokens, similar to Penedo et al. (2023; 2024). We adopt 1.4B parameter models (including embeddings) for our ablation experiments and perform training with a sequence length of 8192, a global batch size of ∼ 1 million tokens, and the StarCoder tokenizer (Li et al., 2023).

**Evaluation:** We evaluate our models using LM Evaluation Harness (Gao et al., 2024) on two categories of tasks: 11 *High-Signal tasks* (18 variants combining 0-shot and few-shot) and 20 *Extended tasks* (29 variants combining 0-shot and few-shot). For ablations analyzing individual ingredients and for tuning thresholds, we evaluate the models on a subset of 8 high-signal tasks to reduce risk of overfitting the filtering thresholds to the evaluation sets. We use extended evaluation set in final evaluations to validate generalization. See Appendix E for details on the benchmarks and Appendix G for details on the experimental setup.

### 2.2 GNEISSWEB RECIPE INGREDIENTS ALONG WITH KEY ABLATIONS

#### 2.2.1 EXACT SUBSTRING DEDUPLICATION

Removing duplicates from training data has been shown to reduce memorization (Kandpal et al., 2022; Carlini et al., 2023) and improve model performance (Lee et al., 2022; Penedo et al., 2023). Although FineWeb applied per snapshot fuzzy deduplication (Penedo et al., 2024) (details in Appendix B), duplicates still remain at sequence-level within and across documents.

---

[2] https://github.com/IBM/data-prep-kit
[3] Publicly released components: https://huggingface.co/datasets/ibm-granite/GneissWeb

We apply exact substring deduplication to remove any substring of predetermined length that repeats verbatim more than once by adapting the implementation from Lee et al. (2022) that is based on Suffix arrays (Manber & Myers, 1993). We make several modifications to the exact substring deduplication implementation from Lee et al. (2022) to run at scale and adapt it to remove duplicates in a sharded manner (details in Appendix C.1).

**Ablation:** As discussed in Penedo et al. (2024), the impact of deduplication is not typically visible for small number of tokens. Thus, we train two 1.4B models each on 350B tokens as follows. The baseline model is trained on 350B tokens randomly sampled from FineWeb-V1.1.0, and the second model is trained on the 350B tokens randomly sampled after applying sharded exact substring deduplication to FineWeb-V1.1.0. In Figure 2, we compare average evaluation score on high-signal tasks for the two models. We see that for both datasets compared, the average score increases as the training progresses, and the score of the model trained on the dataset with exact substring dedu-

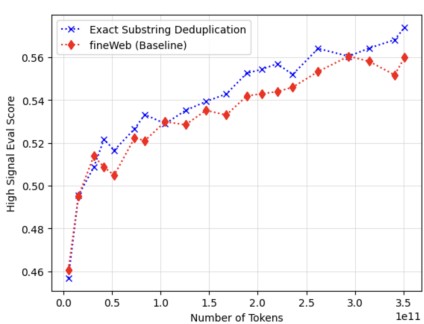

Figure 2: Ablation experiment comparing Exact Substring Deduplication against the FineWeb.V1.1 baseline at 1.4 Billion model size for 350 Billion tokens.

plication is consistently higher (especially after 260B tokens) ending at 57.39 percent than the baseline which ends at 55.99 percent.

### 2.2.2  FASTTEXT QUALITY CLASSIFIERS

FastText (Joulin et al., 2017) family of binary classifiers have been used in prior datasets (Weber et al., 2024; Li et al., 2024) for identifying high-quality pre-training documents. Inspired by their effectiveness and efficiency, we use fastText classifiers for quality annotations.

We employ two fastText classifiers for quality annotations: (i) the fastText classifier from Li et al. (2024) trained on a mix of instruction-formatted data (OpenHermes-2.5 (Teknium, 2023)) and high-scoring posts from ELI5 subreddit (Fan et al., 2019), and (ii) an additial fastText classifier trained on a mix of high-quality synthetic data and data annotated by an LLM for high educational value. Specifically, we use the supervised fastText package from Joulin et al. (2017). We use the default fastText architecture and training hyperparameters from the fastText package except for `wordNgrams`.

Table 1: Ablation for fastText quality classifiers. The last row denotes the fastText component of the GneissWeb ensemble filtering (Section 2.2.6).

| Ensemble | High-Signal |
| --- | --- |
| FineWeb-V1.1.0 | 51.94 |
| DCLM-fastText filter | 52.48 |
| Our fastText filter | 52.30 |
| **DCLM-fastText OR our-fastText** | **52.92** |

We use bigrams, i.e., `wordNgrams = 2`, as bigrams are shown to achieve higher performance (Li et al., 2024). We train the classifier on 400k documents, equality split between positive (i.e., high-quality) and negative (i.e., low-quality) classes, where positive documents are primarily selected from the open synthetic dataset Cosmopedia (Ben Allal et al., 2024) and negative documents are random documents from FineWeb, annotated with Mixtral-8x22B-Instruct (details in Appendix C.2). In Appendix I, we present examples showing the effectiveness of our custom fastText filter.

**Ablation:** We compare a 1.4B model trained on 35B random tokens from FineWeb against a model trained on 35B random tokens from FineWeb with fastText quality filters applied (see Table 1). We observe that our fastText classifier improves the performance and complements DCLM-fastText to achieve further improvements. Here DCLM-fastText OR our-fastText denotes the fastText component of the GneissWeb ensemble filtering rule (details in Section 2.2.6 and Appendix C.6).

### 2.2.3 READABILITY SCORES

Readability scores are formulas based on text statistics (such as sentence length, average number of words, etc.) designed to assess how easily the text can be read and understood (Duffy, 1985). We apply readability scores as a novel quality metric to facilitate identifying and filtering hard-to-read low-quality documents.

We experimented with a number of readability score formulas including Flesch-Kincaid-grade level (Kincaid et al., 1975), Automated Readability Index (ARI) (R.J.Senter & E.A.Smith, 1967), Gunning Fog (Gunning, 1952) and McAlpine-EFLAW (McAlpine, 2006; Mueller, 2012), and determined that McAlpine-EFLAW yields the best results (details in Appendix C.3). McAlpine-EFLAW readability score of a document $D$ is defined as $(W + M)/S$, where $W$ denotes the number of words in $D$, $M$ denotes the number of miniwords (words with 3 or fewer characters) in $D$, and $S$ denotes the number of sentences in $D$. Lower McAlpine-EFLAW readability score indicates the document is easier to understand for a reader with English as a foreign language. Further, we analyzed readability score distributions of the documents grouped by categories (details in Appendix C.3), and observed that distributions of certain categories differ from the overall distribution across categories. These specific categories tend to contain many documents with educational-style content, resulting in higher values of readability scores. Equipped with this observation, we design *category-aware readability score filter* wherein we select lenient filtering threshold on readability scores for documents from these educational-style categories, and stricter filtering threshold for documents outside of these categories. We select initial thresholds based on readability score distributions, and then perform grid search to tune the thresholds. We use lenient thresholds for the following educational-style categories: science, education, technology and computing, and medical health. Further experiments showed that including other categories, e.g., adding "news and politics", "business and finance" and "personal finance" to the hard-to-read categories did not improve the performance. See ablations in Appendix C.3.

**Ablation:** We provide results of ablations on different readability scores used to determine the best readability score that provides the maximum performance gain in Table 2. We see that the model trained on 35B random tokens from FineWeb-V1.1.0 with McAlpine-EFLAW readability score quality filter applied achieves the final score of 53.2% as compared to the score of 51.94% for the baseline model trained on 35B random tokens from FineWeb-V1.1.0. Appendix I presents examples of low-quality documents filtered using the readability score.

Table 2: Comparison of Average Eval Scores on High Signal tasks for different readability-score filters and extreme-tokenized documents filters.

| Ensemble | High-Signal |
| --- | --- |
| FineWeb-V1.1.0 | 51.94 |
| **McAlpine-EFLAW quality filter** | **53.20** |
| Flesch-Kincaid quality filter | 52.05 |
| Automated Readability Index quality filter | 52.32 |
| Gunning Fog quality filter | 52.26 |
| **Extreme-tokenized quality filter** | 52.85 |

### 2.2.4 EXTREME-TOKENIZED DOCUMENTS

On manual inspection of a number of documents, we found some low-quality documents mislabeled by fastText classifiers and the readability score. After tokenizing these documents, we observed a peculiar pattern: while most of the documents have similar lengths, they produced significantly different token counts. To quatify this effect, we propose novel annotations that effectively leverage information from the "pre-tokenization" stage (document char length, document size) and the "post-tokenization" stage (token counts) to identify potential low-quality documents. Specifically, for each document, we compute *TokensPerChar* – the number of tokens divided by the number of characters and *TokensPerByte* – the number of tokens divided by the size in bytes.

We analyzed the distributions of TokensPerChar and TokensPerByte for documents grouped by categories, and observed that low-quality documents typically fall into the two extremes of the distribution (see Appendix C.4 for details). Therefore, we characterize extreme-tokenized documents of a given category as those falling into the two extremes of the TokensPerChar (or TokensPerByte) distribution for the category. Furthermore, distributions of the documents in specific education-style categories differ than the overall distribution across categories. Guided by this observation, we design our *category-aware extreme-tokenized documents filter*, in which, we select lenient thresholds

on TokensPerChar/TokensPerByte for the specific categories and stricter thresholds for the other categories. Specifically, we select lenient thresholds for the same categories as in the case of readability scores: science, education, technology and computing, and medical health. Further experiments show that adding other categories (where distributions differ) such as personal finance degrade performance. We choose initial thresholds based on the distributions, and then perform grid search to tune the thresholds (see ablations in Appendix C.3).

**Ablation:** Table 2 shows the results of the ablation experiment with the best thresholds. We see that the score of the model trained on 35B tokens randomly sampled from FineWeb-V1.1.0 with extreme-tokenized quality filter applied ends at 52.85%, which is higher than 51.94% achieved by the baseline model trained on 35B random tokens from FineWeb-V1.1.0. Appendix I presents examples of low-quality extreme-tokenized documents.

### 2.2.5 DOCUMENT CATEGORY CLASSIFIERS

As mentioned in previous sections, the quality score distributions of documents in certain categories, which tend to contain documents with high educational-level, differ from the overall distribution across all categories. In particular, we observe that the following Interactive Advertising Bureau (IAB) Tech Lab categories (IAB, 2017) supported by WatsonNLP categorization (Team, 2024) have significantly different distributions than the overall distribution across all categories: *science, education, technology & computing, and medical health*. Thus, we annotate whether each document falls into any of these key categories. To perform category classification, we train four binary fastText category classifiers for each of the four key categories (more details in Appendix C.5). We leverage these category annotations in our quality filtering which results in better performance compared to filtering without leveraging category information.

### 2.2.6 ENSEMBLE QUALITY FILTER

Equipped with multiple quality annotators, we develop an ensemble quality filter with the aim of maximizing data quality under the constraint of retaining nearly 10T tokens from FineWeb-V1.1.0. We consider five ensemble aggregation rules described in Appendix D. We tune the thresholds for fastText classifiers for a given ensemble filtering rule such that around 10T tokens are retained from the 15T tokens of FineWeb-V1.1.0. The GneissWeb ensemble filtering rule is described in detail in Figure 6 in Appendix C.6 and the GneissWeb recipe is outlined in Figure 7 in Appendix C.7. We provide explicit thresholds for all our component filters in Table 11 in Appendix C.7. Note that the ensemble rules are invariant to the order of operations for a given set of thresholds (details in Appendix C.4).

**Ablation:** Table 3 shows the average score on high-signal tasks for the five ensemble filtering rules described in Appendix D. We see that the GneissWeb ensemble filtering rule outperforms the other ensemble filtering rules as well as the individual components. To verify whether the gains scale with the model parameters and other tasks, we also perform an ablation by training 7B parameter models trained on 100B tokens. Due to compute restrictions, we focus on the comparison with ensemble filtering rule 1 – the second best rule in

Table 3: Comparison of Average Eval Scores on High Signal tasks for various ensemble filtering rules.

| Ensemble | High-Signal |
|---|---|
| FineWeb-V1.1.0 | 51.94 |
| Ensemble filtering rule 1 | 53.53 |
| Ensemble filtering rule 2 | 52.91 |
| Ensemble filtering rule 3 | 52.79 |
| Ensemble filtering rule 4 | 52.56 |
| **GneissWeb ensemble filtering rule** | **54.29** |

35B ablations. Table 13 in Appendix C.6 shows the average eval score on high-signal tasks as well as extended tasks for the filtering rules along with the baseline of FineWeb-V1.1.0. We observe that the GneissWeb filtering ensemble rule outperforms on both high-signal and extended tasks.

## 3 IMPLEMENTATION AND OPEN SOURCING

We implemented the GneissWeb recipe using an open-source Data Prep Kit library (Wood et al., 2024). Additionally, we also created a faster method to get to an approximation of GneissWeb using *Bloom filters*. In Appendix G.4, we provide resource consumption details.

Table 4: **Comparison of the GneissWeb dataset with other public large datasets.** Average scores of 1.4B parameter models trained on 350B tokens randomly sampled from state-of-the-art open datasets. Scores are averaged over 3 random seeds used for data sampling and are reported along with standard deviations. GneissWeb performs the best among the class of large datasets.

| Dataset | Tokens | High-Signal Eval Score | Extended Eval Score |
|---|---|---|---|
| FineWeb-V1.1.0 | 15T | $56.26 \pm 0.14$ | $47.33 \pm 0.30$ |
| **GneissWeb** | **9.8T** | **$58.40 \pm 0.19$** | **$48.82 \pm 0.27$** |
| FineWeb-Edu-Score-2 | 5.4T | $57.36 \pm 0.42$ | $48.16 \pm 0.29$ |

**Data Prep Kit Transforms:** The kit provides various functions necessary for data processing through an interface, called *transform*. The input to a transform is a collection of documents with annotations including metadata (document id, etc.) and labels given by other transforms, which usually corresponds to a single parquet file[4]. The output is either the same collection of documents with additional annotations such as document quality scores by certain criteria or a subset of the input if the transform performs document filtering. The transforms implemented include Exact Substring Deduplication, Quality Annotation and Category Classification using fastText models, Readability Score Annotator, Extreme-Tokenized-Documents Annotator, and Ensemble Filtering.

**Bloom Filter:** We provide an inexpensive way of reproducing an approximation of GneissWeb by creating a Bloom filter (Bloom, 1970) of the *document ids* of GneissWeb. This filter was created using the rbloom (Hanke, 2023) package with a false positive rate set to 0.0001. Given that GneissWeb has $\sim$12B documents, the bloom filter is of $\sim$28GB in size. One can use either FineWeb or Common Crawl snapshots and probe the Bloom filter with the document ids to determine if a document is in GneissWeb or not (more details in Appendix F). We develop a Data Prep Kit transform which can take a parquet file as input and output the parquet file with an additional boolean column "is-in-GneissWeb" indicating whether the document is in GneissWeb.

**Open Sourced Components:** The Gneissweb recipe notebook[5] provides implementation details of each Data Prep Kit transform along with steps to create the GneissWeb dataset. We have also open sourced[6] the fastText models for quality annotation as well as fastText classifiers for the science, technology & computing, education, and medical health categories. Furthermore, we have open sourced the notebook[7] for the GneissWeb Bloom filter along with the Gneissweb Bloom filter[8].

## 4 EVALUATING THE GNEISSWEB DATASET

**Evaluation Set Up:** We compare GneissWeb with other datasets by training data ablation models that are identical in terms of architecture and training parameters, except for the data they were trained on. The model architectures and setups for training and evaluations are described in Section 2.1. We perform evaluations on 11 *High-Signal tasks* (18 variants combining 0-shot and few-shot) and 20 *Extended tasks* (29 variants combining 0-shot and few-shot) (more details in Appendix E).

In the following experiments comparing our dataset with other open-source datasets, we train the models on 350B tokens. Our experimental scale (100B–350B tokens) aligns with several dataset papers (Penedo et al., 2024; 2023; Soldaini et al., 2024). Furthermore, to minimize the impact of random data subset selection on evaluation scores, we use three equal-sized random subsets of the full data to train three models, and compute average scores along with standard deviation. Following the literature from Penedo et al. (2023; 2024); Li et al. (2024); Soldaini et al. (2024), our experiments are focused on small (1.4B), medium (3B), and large (7B) model sizes. Developing high quality dataset requires an adequate iteration speed for ablation experiments. This makes it difficult to perform experiments on larger number of tokens or larger models, which are expensive in terms of training cost and time. Li et al. (2024) have shown high rank correlation between performance results at smaller scales and those at larger scales, which suggests that our performance gains at 7B models at 350B tokens can transfer to larger models and number of tokens.

---

[4]https://parquet.apache.org

[5]https://github.com/IBM/data-prep-kit/blob/dev/examples/notebooks/GneissWeb/GneissWeb.ipynb

[6]https://huggingface.co/datasets/ibm-granite/GneissWeb

[7]https://github.com/ian-cho/data-prep-kit/blob/dev/transforms/universal/bloom/bloom_python.ipynb

[8]https://huggingface.co/ibm-granite/GneissWeb.bloom

Table 5: **GneissWeb outperforms other large public datasets (5T+ tokens) at 3B and 7B model size.** Average Scores on High Signal and Extended Tasks for 3B and 7B models trained on 350B tokens. Scores are averaged over 3 random seeds used for data sampling and are reported along with standard deviations.

| Dataset | 3B models | | 7B models | |
|---|---|---|---|---|
| | High-Signal tasks | Extended tasks | High-Signal tasks | Extended tasks |
| FineWeb.V1.1.0 | $60.31 \pm 0.21$ | $50.15 \pm 0.07$ | $64.61 \pm 0.23$ | $53.39 \pm 0.25$ |
| **GneissWeb** | $\mathbf{62.83 \pm 0.24}$ | $\mathbf{52.10 \pm 0.22}$ | $\mathbf{67.34 \pm 0.26}$ | $\mathbf{55.14 \pm 0.28}$ |
| FineWeb-Edu-Score-2 | $61.63 \pm 0.04$ | $51.13 \pm 0.17$ | $65.51 \pm 0.34$ | $54.61 \pm 0.31$ |

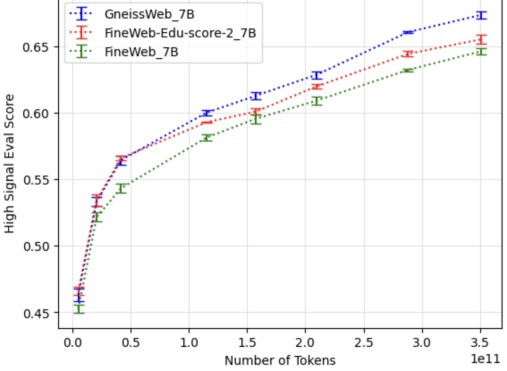
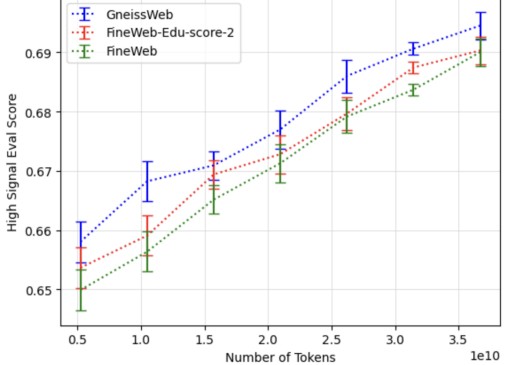

Figure 3: Average evaluation score on High-Signal tasks versus the number of tokens for 7B parameter models. The models trained on GneissWeb consistently outperform the ones trained on FineWeb.V1.1.0 and FineWeb-Edu-score-2.

Figure 4: Average evaluation score on High-Signal tasks versus the number of tokens for Stage-2 pre-training. Scores are averaged over 3 random seeds used for data sampling and are reported along with standard deviations.

We compare GneissWeb with FineWeb[9] (15T tokens) and FineWeb-Edu-Score-2 (5.4T tokens) (Penedo et al., 2024). because these are widely regarded as state-of-the-art open datasets for Stage-1 training. They meet the 5T+ token threshold and have demonstrated superior performance over several alternatives (Appendix B). In Appendix H, we provide additional experiments comparing GneissWeb against RedPajamaV2 (Weber et al., 2024), TxT-360 (Tang et al., 2024), and Dolma Soldaini et al. (2024), and open models.

**1.4B Models Trained on 350B Tokens:** Table 4 shows the average scores on high-signal tasks and extended tasks for 1.4B parameter models trained on three randomly sampled sets of 350B tokens from each dataset. Models trained on GneissWeb outperform those trained on FineWeb-V1.1.0 by 2.14 percent points on high-signal tasks, and by 1.49 percent points on extended tasks. Models trained GneissWeb also outperform those trained on FineWeb-Edu-Score-2.

When the performance is broken down into the various categories of tasks – Commonsense Reasoning, Language Understanding, Reading Comprehension, World Knowledge, and Symbolic Problem Solving, GneissWeb is not only the best overall among the datasets that are greater than 5T token set size, but in fact performs the best in all categories of tasks except World Knowledge (see Table 17 in Appendix H).

**3B and 7B Models Trained on 350B Tokens:** To evaluate GneissWeb for training larger models, we train models with 3B and 7B parameters on three independent sets of 350B tokens sampled randomly from datasets. Table 5 depict the evaluation scores for the 3B and 7B models. We observe that performance gains of GneissWeb improve for large models. Models trained on GneissWeb outperform those trained on FineWeb.V1.1.0 by 2.52 percent points for 3B model and 2.73 percent points for 7B model in terms of the average score computed on high-signal benchmarks. GneissWeb outperforms FineWeb V1.1.0 by 1.95 percent points for 3B model and 1.75 percent point for 7B model on Extended benchmarks. Figure 3 shows that GneissWeb demonstrate steeper scaling laws

---

[9]We used FineWeb-V1.1.0 `https://huggingface.co/datasets/HuggingFaceFW/fineweb`

Table 6: Fairness, bias and toxicity evaluation of 7B models trained on large datasets.

| Dataset | Winogender (Fairness, ↑) | CrowS-Pairs (Bias, → 0.5) | Real Toxicity Prompts (Toxicity, ↓) |
|---------|--------------------------|----------------------------|--------------------------------------|
| FineWeb-V1.1.0 | $60.69 \pm 1.82$ | $0.66 \pm 0.012$ | 0.00 |
| FineWeb-Edu-Score-2 | $59.86 \pm 1.83$ | $0.67 \pm 0.012$ | 0.00 |
| **GneissWeb** | $59.58 \pm 1.80$ | $0.68 \pm 0.011$ | 0.00 |

than the alternatives, with consistently higher evaluation score. Similar results are observed for the 1.4B and 3B models (see Figures 10 and 11 in Appendix H).

**Stage-2 Pre-training Evaluation Results:** We evaluate model performance when Stage-2 pre-training is performed with a smaller, higher quality dataset (such as FineWeb-Edu (Penedo et al., 2024) or DCLM-Baseline (Li et al., 2024)). We start with three checkpoints of the 7B model, each trained on random 350B tokens from three Stage-1 pre-training datasets: FineWeb V1.1.0, FineWeb-Edu-Score2, and GneissWeb. We then continue training each checkpoint on 35B tokens sampled randomly from a Stage-2 pre-training dataset, DCLM-Baseline. Figure 4 shows that the GneissWeb model continues to demonstrate steeper scaling laws than the alternatives, with consistently higher evaluation score. This ablation shows that the performance gain achieved by GneissWeb models in Stage-1 continues in Stage-2 pre-training when higher quality dataset is used.

**Fairness, Bias, and Toxicity:** We extended our evaluation suite to include the following benchmarks: Winogender (Rudinger et al., 2018), Crows-Pairs (Nangia et al., 2020), and Real Toxicity Prompts (Gehman et al., 2020). The performance of 7B ablation models trained on FineWeb, FineWeb-Edu-Score-2, and GneissWeb are given in Table 6. We briefly review these benchmarks: Winogender (higher is better) measures how likely a model is to reinforce a gender-based stereotype when infilling a gendered pronoun. CrowS-Pairs (lower is better; ideal is 0.5) measures bias score of a model as the percentage of stereotypical sentences that are rated as more likely by the model than the non-stereotypical sentences. Real Toxicity Prompts (lower is better) measures how easily a user can prompt a model to generate toxic content. Since Real Toxicity Prompts evaluations are much slower, we restricted to 16000 prompts. The results on 7B ablation models show that models trained on GneissWeb perform comparably on the fairness, bias, and toxicity benchmarks, indicating that GneissWeb does not introduce disproportionate risks in these areas.

**Training time efficiency of GneissWeb:** Our GneissWeb recipe employs judiciously constructed quality filters to retain "high quality" tokens from FineWeb. The improved token quality results in significant efficiency gains during pre-training. We compute efficiency gains by estimating the number of training FLOPs for achieving a target evaluation performance. We choose the average high-signal eval score of 64% for 7B parameter models (since

Table 7: Training FLOPs to achieve the high-signal eval score of 64% for 7B models.

| Dataset | Train FLOPs |
|---------|-------------|
| FineWeb-V1.1.0 | $2.2 \times 10^{21}$ |
| FineWeb-Edu-Score-2 | $1.8 \times 10^{21}$ |
| **GniessWeb** | $\mathbf{1.8 \times 10^{21}}$ |

FineWeb performance plateaus around this; see Figure 3). Table 7 shows the FLOPs to achieve the targer score, computed using the transformer FLOP estimation from Li et al. (2024). We observe that GneissWeb achieves the same performance with 27% smaller number of FLOPs than FineWeb, achieving higher training efficiency, thereby reducing compute costs.

**Adaptability on downstream datasets:** We study the performance of models pre-trained on GneissWeb on downstream tasks. Since the key goal of instruction tuning is to enable LLMs to follow instructions, we focus on the task of instruction following. We take 7B models trained on FineWeb-Edu-Score-2 and GneissWeb, and perform supervised fine-tuning with a subset of Tulu3-SFT-mix (Lambert et al., 2025). We evaluate the models on IFEval Zhou et al. (2023) – a benchmark that measures the instruction following ability of models. The prompt-level (strict) accuracy measures the percentage of prompts for which the model strictly followed all instructions in the prompt. The model trained on GneissWeb achieves the accuracy of 25.69 percent, whereas the one trained on FineWeb-Edu-Score-2 achieves 23.84 percent accuracy on IFEval. This demonstrates the adaptability of models trained on GneissWeb to downstream tasks.

## 5 CONCLUSION AND LIMITATIONS

We introduced the GneissWeb recipe and demonstrated how to improve upon state-of-the-art datasets of similar size, achieving a better trade-off between data quality and quantity. The key differentiators of the GneissWeb recipe included novel category-aware extreme-tokenized documents quality filter and category-aware quality filter based on human readabilty, along with a judiciously constructed ensemble of filters.

Similar to several prominent open datasets in the literature, GneissWeb focuses mainly on English data. More work is needed to adapt our processing steps and the GneissWeb recipe to multilingual datasets. We performed our ablation experiments with only one tokenizer (StarCoder), and other tokenizers may perform better, especially on multilingual or math data. The focus of filtering steps is on language quality and it is likely that code and math content is limited. GneissWeb can be augmented with code and math data sources to improve the performance on code and math related tasks.

## REPRODUCIBILITY STATEMENT

We provide key details of the GneissWeb recipe in the main paper, and full details in Appendix C. We present the formal algorithm for the GneissWeb ensemble filter in Figure 6 in Appendix C.6 and exact thresholds for all filters used in the GneissWeb recipe in Appendix C.7. A Jupyter notebook for the GneissWeb recipe, covering all the details, is attached as a supplemental material. We will open source fastText classifiers trained for GneissWeb on Hugging Face. We will also open source the GneissWeb recipe notebook, code for all processing steps, ablations models, and the Bloom filter that we created for efficient reproduction of GneissWeb from the already open-sourced FineWeb dataset. For ablation and dataset comparison experiments, we provide details in Appendix D, the list of evaluation benchmarks (with references) in Appendix E, and details of the Bloom filter in Appendix F. We present details of the model architectures and hyperparameters used in our experiments in Appendices G.2 and G.3, respectively. We also outline the compute infrastructure used in our experiments in Appendix G.1.

## ETHICS STATEMENT

Our starting point is FineWeb (Penedo et al., 2024), which applied URL filtering to remove adult content. Furthermore, FineWeb applied Personal Identifiable Information (PII) removal, by anonymizing email and public IP addresses. FineWeb also performed bias analysis to demonstrate that biases across the gender, religion, and age subgroups in the dataset are not strong (details in Penedo et al. (2024)). We note that our GneissWeb is a subset of FineWeb, no new data is added. To analyze the impact of the GneissWeb recipe on fairness, bias, and toxicity, we extend our evaluation suite as discussed in Section 4, Table 6. While our experiments indicate minimal negative impact, we acknowledge that our evaluations are far from complete.

## ACKNOWLEDGEMENT

We would like to acknowledge the efforts of numerous teams at IBM Research AI and Hybrid Cloud Platform, IBM AI Infrastructure, IBM Software, IBM Data and Model Governance, the IBM Brand, Marketing and Communications teams. We would like to specially thank Shahrokh Daijavad, Yohei Ikawa, and Yang Zhao for their contributions to Data Prep Kit and Bloom filters. Additionally, we would like to thank IBM Research leaders - Dario Gil, Sriram Raghavan, Mukhesh Khare, David Cox for their support. We would like to thank and acknowledge the insightful feedback from Dakshi Agrawal, Heiko Ludwig and Rameswar Panda as well as the help provided by Basel Shbita, Chirag Garg, Jay Pankaj Gala and Pengyuan Li for data downloads and experiments.

We would also like to acknowledge the creators of FineWeb from HuggingFace, creators of fastText from Facebook AI Research (FAIR) lab, and creators of DCLM-FasText from ML Foundations, we used FineWeb as base dataset for GneissWeb and trained fastext style filters used to produce GneissWeb.

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

## A  RELATED WORK

In this work we aim to create a large dataset capable for pre-training of a LLM. There are several related works in this space. Prior public pre-training datasets are typically derived from the Common Crawl (Crawl, 2007). Early works include the C4 dataset with 160 billion tokens (Raffel et al., 2020) and the Pile dataset with billion tokens Gao et al. (2020). The C4 dataset is curated from the April 2009 snapshot of the Common Crawl. It uses langdetect (Library, 2014) to detect English text, applies a series of heuristic filters including discarding any page with less than 3 sentences, removing lines without any terminal punctuation mark, removing any page containing any word in a list of dirty, naughty, obscene or bad words etc, and also performs deduplication by removing all but one of any three-sentence span occurring more than once in the dataset. The Pile is a composite dataset that includes the Pile-CC, which is based on Common Crawl. It uses pycld2 (pycld2 Library, 2019) for language detection, removes boilerplate using jusText (Library, 2024a), applies classifier-based filtering and performs fuzzy deduplication.

Multilingual models like XLM RoBERTa (Conneau et al., 2020) used the CC100 dataset (Conneau et al., 2019). This dataset was curated using the CCNET (Wenzek et al., 2019) processing pipeline on one year of Common Crawl snapshots. CCNet uses the data processing methods introduced in fastText (Joulin et al., 2017), which include deduplicating documents and applying LangID filtering. It then adds a filtering step to select documents that are similar to high-quality corpora like Wikipedia by utilizing a 5-gram KenLM filter.

RedPajama dataset (Weber et al., 2024) is an open source attempt to recreate the dataset used to train Llama models. It is a composite dataset which includes text obtained from the Common Crawl by using the CCNet pipeline (Wenzek et al., 2019) and a classifier trained to identify documents similar to Wikipedia articles or references. SlimPajama with 627B tokens Soboleva et al. (2023) further refines RedPajama by removing short documents and performing additional fuzzy dedupllication. RedPajama-V2 (Weber et al., 2024) with 30 trillion tokens is entirely based on the Common Crawl and contains annotations without applying any filtering. These annotations cover filtering techniques from CCNet, C4, and others, and also labels identifying deduplicates using exact and fuzzy deduplication.

RefinedWeb dataset (Penedo et al., 2023) is a Common Crawl-based dataset, using trafilatura (Library, 2024b) for text extraction, fastText-based language identification (Joulin et al., 2017), heuristic rules for quality filtering, and fuzzy and exact deduplication. Dolma (Soldaini et al., 2024) is a 3 trillion token composite dataset with a Common Crawl-based portion, which employs fastText for language identification, primarily uses heuristic rules from MassiveWeb (Rae et al., 2022) for quality filtering, applies toxicity filtering based on rules and classifiers and performs deduplication at URL, document and paragraph levels.

More recent datasets include FineWeb datasets (Penedo et al., 2024), DCLM-Baseline (Li et al., 2024), and TxT360 (Tang et al., 2024). FineWeb consists of 15T tokens derived from the Common Crawl by applying a series of processing steps, mainly including language classification, fuzzy deduplication at snapshot level and heuristic rule-based quality filters. Subsequently, two smaller but higher quality versions called FineWeb-Edu (1.3 trillion tokens) and FineWeb-Edu-Score2 (5.4 trillion tokens) derived from FineWeb were released (Penedo et al., 2024). These smaller high quality derivatives of FineWeb are created by retaining documents perceived to have higher educational value from FineWeb (see Appendix B for more details).

DCLM-Baseline (3.8 trilion tokens) is obtained from the Common Crawl snapshots by using resiliparse (Library, 2021) for text extraction, heuristic quality filters from RefinedWeb, fuzzy deduplication with Bloom filter (Groeneveld, 2023), model-based quality filtering using a specially trained fastText classifier. TxT360 is a composite dataset obtained from Common Crawl snapshots and 14 high-quality datasets (e.g. FreeLaw, Ubuntu IRC, etc). TxT360 is obtained by first applying local exact deduplication, global fuzzy deduplication, and quality filtering to both web and curated datasets, resulting in approximately 5 trillion tokens, which are then up-sampled to over 15 trillion tokens. The mixing and up-sampling approach is shown essential to boosting TxT360 performance.

Nemotron-CC (Su et al., 2024) and Zyda2 (Tokpanov et al., 2024) are two of the most recent works. Zyda-2 is a 5T high-quality token dataset obtained by collating high-quality open-source datasets including FineWeb-Edu, DCLM, Zyda-1, and Dolma-CC and then applying cross-deduplication and

model-based quality filtering. Nemotron-CC is a 6.3T token dataset, including 4.4 trillion tokens from Common Crawl by applying exact substring deduplication, global fuzzy deduplication and model-based quality filtering. Nemotron-CC also includes 1.9T synthetic tokens (approximately 30% of the data) generated using a rephrasing-based approach from low-quality and high-quality documents. Adding synthetic data (like Nemotrcon-cc) to GneissWeb and combining portions of GneissWeb to already curated datasets (like Zyda-2) will further improve the quality.

Our focus is on datasets with more than 5 trillion tokens, derived entirely from Common Crawl. The token requirement is motivated from the long token horizon Stage-1 pre-training requirements of LLMs. We take FineWeb (Penedo et al., 2024) as the starting point to build our dataset since FineWeb is sufficiently large dataset with 15T tokens which has been shown to outperform several public datasets – C4, RefinedWeb, Dolma, RedPajamaV, SlimPajama and the Pile. While FineWeb-Edu, FineWeb-Edu-Score-2 (Penedo et al., 2024) and the recent DCLM-Baseline (Li et al., 2024) improve data quality over FineWeb they do so by performing aggressive model-based quality filtering. Such an aggressive filtering cuts down their size which may not be sufficient for pre-training (as pre-training typically consists of only one pass or few passes over the pre-training dataset (Muennighoff et al., 2023)). Our GneissWeb recipe achieves a favorable trade-off between data quality and quantity thereby producing ∼10T high quality tokens with higher performance than prior datasets with 5T+ tokens.

## B  FINEWEB DATASETS

FineWeb (Penedo et al., 2024) is obtained from the Common Crawl (CC) (Crawl, 2007) by applying the following processing steps.

1. Text is extracted from the CC WARC (Web ARChive format) files using trafilatura (Library, 2024b).

2. *Base filtering* is applied on the text file consisting of the following steps: URL filtering using a blocklist to remove adult content, fastText language classifier (Joulin et al., 2017) to keep English documents with a score of at least 0.65, and quality and repetition removal filters from MassiveText Rae et al. (2022).

3. Fuzzy deduplication is performed on each individual CC snapshot using the MinHash algorithm (Broder, 1997).

4. All the heuristic quality filters from the C4 dataset (Raffel et al., 2020) are applied, except for the terminal punctuation filter (retaining only those lines that end in a terminal punctuation mark).

5. Three additional heuristic filters are applied: remove documents where the fraction of lines ending with punctuation is $<= 0.12$, where the fraction of characters in duplicated lines is $>= 0.1$, and/or where the fraction of lines shorter than 30 characters is $>= 0.67$.

FineWeb-Edu is obtained by applying an educational quality classifier developed from synthetic annotations generated by Llama-3-70B-Instruct (Grattafiori et al., 2024). FineWeb-Edu uses a higher educational score threshold of 3 to retain 1.3T tokens, and FineWeb-Edu-Score-2 uses a lower educational score threshold of 2 to retain 5.4T tokens. We take FineWeb as the starting point to build our dataset since FineWeb is a sufficiently large dataset with 15T tokens which has been shown to outperform several public datasets — C4, RefinedWeb, Dolma, RedPajamaV, SlimPajama and the Pile as shown by Penedo et al. (2024).

## C  THE GNEISSWEB RECIPE

In this section we provide details of individual components of the GneissWeb recipe.

### C.1  EXACT SUBSTRING DEDUPLICATION

Removing duplicates from training data has been shown to reduce memorization (Kandpal et al., 2022; Carlini et al., 2023) and improve model performance (Lee et al., 2022; Penedo et al., 2023). FineWeb applied per snapshot fuzzy deduplication and removed near-duplicate documents using

the MinHash algorithm (Penedo et al., 2024). Furthermore, FineWeb also applied repetition filter, intra-document deduplication (Rae et al., 2022) which removes documents with many repeated lines and paragraphs. (See Appendix B for details on FineWeb.) However, duplicates still remain at sequence-level within and across documents. Such repeated substrings bypass the *document level* deduplication steps of FineWeb for several reasons: they may not represent a significant enough portion of a document or a single document may include repeated sections from various documents.

We apply exact substring deduplication to remove any substring of predetermined length that repeats verbatim more than once by adapting the implementation from Lee et al. (2022) based on Suffix arrays (Manber & Myers, 1993). Exact substring deduplication can be fine tuned through two hyper-parameters: length-threshold (the minimum length of repeated text sequences) and frequency-threshold. We utilize a length-threshold of 50, consistent with the implementation from Lee et al. (2022); Penedo et al. (2023).

We make several modifications to the exact substring deduplication implementation from Lee et al. (2022) to run at scale. Furthermore, we adapt it to remove exact substring duplicates in a sharded manner. In particular, we shard each snapshot of FineWeb-V1.1.0 into sets of roughly equal size and apply exact substring deduplication on each shard independently. Also, rather than removing all copies of a duplicate substring, we retain the first occurrence of each duplicate substring and remove any subsequent matches exceeding 50 consecutive tokens.

## C.2 FASTTEXT QUALITY CLASSIFIERS

FastText (Joulin et al., 2017) family of binary classifiers have been used in prior datasets (Weber et al., 2024; Li et al., 2024) for identifying high-quality pre-training documents. Recently, (Li et al., 2024) showed that fastText classifier trained on carefully selected data can outperform sophisticated model-based filtering approaches such as AskLLM (prompting an LLM to ask if a document is helpful). Inspired by their effectiveness coupled with the computational efficiency of fastText classifiers, we use fastText classifiers for quality annotations.

We employ two fastText classifiers: (i) the fastText classifier from Li et al. (2024) trained on a mix of instruction-formatted data (OpenHermes-2.5 (Teknium, 2023)) and high-scoring posts from ELI5 subreddit (Fan et al., 2019) and (ii) our own fastText classifier trained on a mix of high-quality synthetic data and data annotated by an LLM for high educational value.

Specifically, we use the supervised fastText package from Joulin et al. (2017) to train a classifier on 400k documents, equality split between positive (i.e., high-quality) and negative (i.e., low-quality) classes, selected as follows.

- Positive documents:

  - 190k synthetic documents randomly sampled from the Cosmopedia dataset – an open synthetic dataset consisting of textbooks, blogposts, stories, posts and WikiHow articles generated by Mixtral-8x7B-Instruct-v0.1 (Ben Allal et al., 2024).
  - 10k documents with high educational value selected as follows: we annotated 600k random documents from FineWeb-V1.1.0 asking Mixtral-8x22B-Instruct to score each document between 1 to 5 for its educational quality (with 5 being the highest quality), using a prompt similar to the one used by FineWeb-Edu. Next, we selected 10k random documents from the documents with scores $\geq 4$.

- Negative documents: 200k random documents out of the 600k Mixtral-annotated documents with scores $\leq 2$.

We denote the DCLM-fastText as $\phi_{\text{DCLM}}$ and our custom fastText as $\phi_{\text{Cosmo}}$. Each fastText classifier takes as input a document $D$ and produces a confidence score between $[0, 1]$ for the document to have positive label (i.e., high-quality).[10] In Appendix I, we present several examples showing how our custom fastText filter complements the DCLM-fastText filter.

---

[10]A fastText classifier conventionally outputs a label (positive or negative) along with the confidence score which can be easily converted to obtain the confidence score for the positive label.

## C.3 READABILITY SCORES

Readability scores are formulas based on text statistics (such as sentence length, average number of words, number of syllables etc.) designed to assess how easily the text can be read and understood (Duffy, 1985). We apply readability scores as a novel quality metric to facilitate identifying and filtering hard-to-read low-quality documents.

A large number of readability score formulas have been developed to asses text difficulty (Sarin & Garraffa, 2023; Begeny & Greene, 2014). We experimented with a number of readability score formulas and selected McAlpine-EFLAW readability score (McAlpine, 2006; Mueller, 2012). McAlpine-EFLAW readability score of a document is a numerical score computed as a function of the number of words in a document plus the number of mini-words (consisting of $\leq 3$ characters) divided by the number of sentences. Lower score indicates the document is easier to understand for a reader with English as a foreign language. Unlike other readability score formulas (such as Flesch-Kincaid (Kincaid et al., 1975) or Gunning Fog (Gunning, 1952)) which are restricted to estimate a grade level for the text, McAlpine-EFLAW produces a numerical score assessing readability for a global audience (Sarin & Garraffa, 2023), making it more suitable for document quality annotation. We also demonstrate the effectiveness of the McAlpine-EFLAW score compared to other readability scores through ablation experiments. Specifically, we tested a few of readability score metrics including Flesch-Kincaid-grade level (Kincaid et al., 1975), Automated Readability Index (ARI) (R.J.Senter & E.A.Smith, 1967), Gunning Fog (Gunning, 1952) and McAlpine-EFLAW, and determined that McAlpine-EFLAW yields the best results.

We analyzed readability score distributions of the documents grouped by categories. Specifically, we considered the documents from the following 3 snapshots from FineWeb-V1.1.0: CC-MAIN-2024-10, CC-MAIN-2023-40 and CC-MAIN-2023-14 and computed the top-level category for each document using the WatsonNLP hierarchical text categorization (Team, 2024). The WatsonNLP categorization is based on the Interactive Advertising Bureau (IAB) Tech Lab categories taxonomy (IAB, 2017). We observe that the distributions are generally bell-shaped for each category, but the values of the mean and variance differ by category. For example, McAlpine-EFLAW readability score distribution in Science has a mean of 27.8 and a standard deviation of 7.57, and McAlpine-EFLAW readability score distribution in Children's TV has a mean of 21.5 and a standard deviation of 7.39. This variation in distributions can be attributed to the observation that several documents in certain categories, such as science, education, technology and medical health demand a higher level of education to understand and have high readability score (higher the readability score, more difficult is the English document to read), leading to a higher average readability score.

Based on this observation, there is a risk of losing high-quality documents if a threshold is selected based on the overall data distribution and the same threshold is applied to all documents. Guided by readability score distributions in different categories, we leverage the category information of documents and develop a category-aware readability score quality filter as part of our ensemble quality filter (see Section 2.2.6 and Appendix C.6 for more details). In general, we use a more lenient threshold for these specific categories to prevent filtering out documents with potential educational value solely because of their high readability scores which results in better performance compared to filtering without leveraging category information. We also performed ablations with other categories. For example, adding "news and politics", "business and finance" as well as "personal finance" to the hard to read categories degraded performance. In Appendix I, we present several low quality examples detected and filtered out by our category-aware readability score filter.

**Thresholds for Readability Score Filter:** We performed ablations on different readability thresholds to identify the configuration that maximizes performance. Models were trained on 35B random tokens filtered using varying thresholds for four key categories and other categories. Evaluation was conducted on high-signal tasks (same setup as Section 2.1). Results in Table 8 show that thresholds of 70 (key categories) and 30 (other categories) achieve the best performance gain.

**Category-Aware Filtering:** Analysis of readability score distributions across document categories demonstrate that distributions of certain categories differ from the overall distribution across categories. These specific categories tend to contain many documents with educational-style content, resulting in different scores than other categories. If a single global threshold is selected based on the overall score distribution and the same threshold is applied to all documents, there is a risk of losing high-quality documents (false-negatives). If the threshold is made lower to avoid potential loss of

Table 8: Readability Score Filter Thresholds

| Threshold for key categories | Threshold for other categories | High-Signal Eval Score |
|---|---|---|
| **70** | **30** | **53.20** |
| 70 | 17 | 52.89 |
| 100 | 25 | 52.67 |
| 70 | 40 | 52.35 |

Table 9: Ablation results for category-aware filtering with redability scores.

| Filter Type | High Signal Eval Score |
|---|---|
| Readability score quality filter: single threshold for all categories | 52.35 |
| Category-aware readability score filter: lenient threshold for 4 key categories ("science", "education", "technology and computing", and "medical health") | **53.20** |
| Category-aware readability score filter: lenient threshold for 7 categories ("science", "education", "technology and computing", "medical health", "news and politics", "business and finance", and "personal finance") | 52.67 |

high-quality documents, then there is a risk of retaining lower quality documents (false-positives). To mitigate this issue, we introduced category-aware thresholds. Category-aware thresholds help us preserve niche but valuable documents.

We provide in Table 9 results of ablations comparing single-threshold filtering versus category-aware filtering. Category-aware filtering improves performance compared to a global threshold. Expanding lenient thresholds beyond the 4 key categories (e.g., adding "news and politics", "business and finance") did not yield further gains, suggesting that our approach effectively balances quality retention and semantic coverage. Observations for extreme-tokenized heuristics are similar: category-aware filtering improved performance compared to a single global threshold, and adding more categories to 4 key categories did not yield further gains.

## C.4 EXTREME-TOKENIZED DOCUMENTS

After manually inspecting fastText model-quality annotations and readability scores of large number of low-quality documents, we found that several abnormal documents were mislabeled by these annotators. We observed a peculiar pattern after tokenizing these documents: while most of these documents had similar lengths, they produced significantly different token counts. To quantify this effect, we propose novel annotations that effectively leverages information from the "pre-tokenization" stage (document char length, document size) and the "post-tokenization" stage (token counts) to identify potential low-quality documents.

Specifically, for each document $D$, we compute the the following two annotations:

$$\text{TokensPerChar}(D) = \frac{\text{Number of Tokens in } D}{\text{Number of Characters in } D}$$

$$\text{TokensPerByte}(D) = \frac{\text{Number of Tokens in } D}{\text{Size of } D \text{ (in bytes)}}$$

We refer to the the documents with extremely high or low number of tokens per character (or tokens per byte) as *extreme-tokenized* documents (see Figure 5 for a schematic).

Data quality filtering based on tokenized data has been used in other works (Mehta et al., 2024; Soldaini et al., 2024) to improve the data quality by filtering out documents with too few tokens (Soldaini et al., 2024) or removing the sequences containing fewer tokens than a specified threshold. However, the effectiveness of these approaches in detecting low-quality documents is limited because of their sole reliance on the token count. Our extreme-tokenized quality filter does not solely rely on token count but also effectively leverages both information from the "pre-tokenization" stage and the "post-tokenization" stage to identify and filter out low-quality documents.

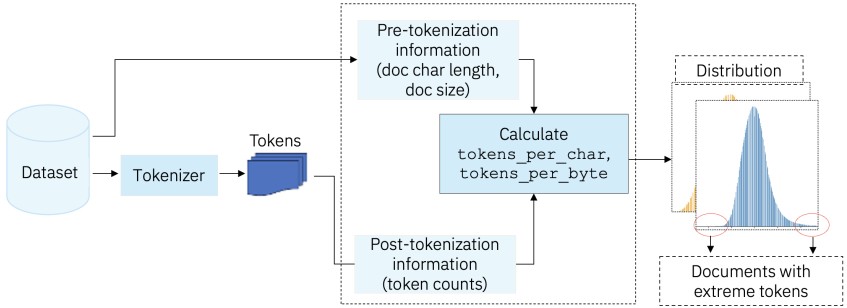

Figure 5: Sequence of steps for removing extreme tokenized documents.

Table 10: Extreme-Tokenized Documents Filter Thresholds

| Lower Bound (key/other) | Upper Bound (key/other) | High-Signal Eval Score |
|---|---|---|
| **0.1/0.22** | **0.5/0.28** | **52.85** |
| 0.15/.02 | 0.45/0.3 | 52.79 |
| 0.17/0.2 | 0.4/0.32 | 52.31 |

We analyzed the distributions of TokensPerChar and TokensPerByte for documents grouped by category. Specifically, we considered the documents from the following 3 snapshots from FineWeb-V1.1.0: CC-MAIN-2024-10, CC-MAIN-2023-40 and CC-MAIN-2023-14, and computed the top-level category for each document using the WatsonNLP hierarchical text categorization (Team, 2024), which is based on the Interactive Advertising Bureau (IAB) Tech Lab categories taxonomy (IAB, 2017). We observe that the distributions are generally bell-shaped for each category, but the values of the mean and variance differ by category. For example, TokensPerChar distribution in Technology & Computing has a mean of 0.22 and a standard deviation of 0.02, and TokensPerChar distribution in Children's TV has a mean of 0.29 and a standard deviation of 0.03. TokensPerByte distribution in Technology & Computing has a mean of 0.22 and a standard deviation of 0.02, and TokensPerChar distribution in Children's TV has a mean of 0.29 and a standard deviation of 0.03. Furthermore, we observe that low-quality documents typically fall into the two extremes of the distribution. Therefore, we characterize extreme-tokenized documents of a given category as those falling into the two extremes of the TokensPerChar (or TokensPerByte) distribution for the category. Guided by the distributions of TokensPerChar and TokensPerByte in different categories, we leverage the category information of documents and develop a category-aware extreme-tokenized quality filter as part of our ensemble quality filter (more details in Section 2.2.6 and Appendix C.6). At a high level, we use stricter thresholds on TokensPerChar/TokensPerByte for documents outside the key categories and use more lenient thresholds for documents in these key categories. In Appendix I, we present several low quality examples detected and filtered out by our category-aware Extreme-Tokenized documents filter.

**Thresholds for Extreme-Tokenized Documents Filter:** We performed ablations on different thresholds to determine optimal bounds. Models were trained on 35B random tokens filtered using varying thresholds for four key categories and other categories. Evaluation was conducted on high-signal tasks (same setup as Section 2.1). Best performance was achieved with bounds (0.1, 0.5) and (0.22, 0.28) for key and other categories, respectively (Table 10).

## C.5 DOCUMENT CATEGORY CLASSIFIERS

As mentioned in previous sections, the quality score distributions of documents in certain categories, which tend to contain documents with high educational-level, differ from the overall distribution across all categories. In particular, we observe that the following IAB categories (IAB, 2017) supported by WatsonNLP categorization have significantly different distributions than the overall distribution across all categories: science, education, technology & computing, and medical health. Thus, for each of these key categories, we annotate whether each document falls into the category.

To perform category classification on the 96 snapshots in FineWeb-V1.1.0 at scale, we train four binary fastText category classifiers for each of the four key categories. Specifically, we generated labeled data using the WatsonNLP hierarchical categorization (Team, 2024), and used the supervised fastText package from Joulin et al. (2017) to train the fastText classifiers on the following documents:

- Positive documents: 400k documents randomly sampled from the documents labeled with that specific category with a confidence score 0.95 and above.

- Negative documents: 400k documents randomly sampled from the documents labeled with any category other than these four categories with a confidence score of 0.95 and above.

We denote the fastText classifiers as $\phi_{sci}$, $\phi_{edu}$, $\phi_{tech}$, and $\phi_{med}$. Each classifier takes as input a document and produces a label whether the document belongs to the category, along with a confidence score between $[0, 1]$.

We use our trained document category classifiers to annotate all the snapshots from FineWeb-V1.1.0. We leverage these category annotations in our category-aware readability score quality filtering and extreme-tokenized quality filtering which results in better performance compared to filtering without leveraging category information.

## C.6 ENSEMBLE QUALITY FILTER

Equipped with multiple quality annotators, we develop an ensemble quality filter with the aim of maximizing data quality under the constraint of retaining nearly 10T tokens from FineWeb-V1.1.0. We construct our ensemble quality filter by selecting thresholds for individual annotators and then designing an ensemble filtering rule for aggregating the filter outputs.

Specifically, we select the thresholds on readability scores integrating the category annotations to design Category-Aware Readability Score filter. We choose our initial thresholds based on the readability score distributions for key categories (computed on entire FineWeb-V1.1.0), and subsequently tune them via grid search to maximize performance gains. Similarly, we select the thresholds for Category-Aware Extreme-Tokenized Documents filter. Then, given an aggregation rule, we choose the thresholds for fastText filters such that we retain nearly 10T tokes from FineWeb-V1.1.0. As an example, a simple aggregation rule is to apply each filter sequentially (which essentially is a logical AND of filter outputs).

We perform ablations on a variety of aggregation rules and determine the *best* aggregation rule that provides the maximum performance gain (see Appendix D for more details). We provide the details of our ensemble quality filter in Figure 6. For the category-aware extreme-tokenized documents filter, we only used TokensPerChar heuristic for our final recipe, as both TokensPerByte and TokensPerChar showed similar distributions.

We provide in detail various ablation experiments in evaluating the impact of our ensemble based filtering rule in Appendix D. We specify explicit thresholds used in the GneissWeb recipe in Appendix C.7.

## C.7 PUTTING IT ALL TOGETHER

The GneissWeb recipe consists of first applying the exact substring deduplication, computing category and quality annotations, and then applying the ensemble quality filter as shown in Figure 7. We obtain the GneissWeb dataset of 10T tokens by applying the GneissWeb recipe to the 15T tokens in the 96 snapshots of FineWeb-V1.1.0.

We specify the exact thresholds used in the GneissWeb recipe in the following.

We note that, while the GneissWeb recipe is designed with the goal of obtaining ∼10T high quality tokens suitable for Stage-1 pre-training, it is also possible to adapt the recipe by tuning filtering parameters to produce smaller and higher quality datasets fit for Stage-2 type of pre-training.

Inputs: Dataset $\mathcal{D}$, Category fastText classifiers $\phi_{\text{sci}}, \phi_{\text{edu}}, \phi_{\text{med}}, \phi_{\text{tech}}$, Readability Score Function Readability and thresholds $\{r_c : c \in \{\text{sci}, \text{edu}, \text{tech}, \text{med}\}\}$, and extreme-tokenized threshold tuples $\{(\tau_c^{\text{Low}}, \tau_c^{\text{High}}) : c \in \{\text{sci}, \text{edu}, \text{tech}, \text{med}, \text{other}\}\}$, fastText annotators $\phi_{\text{DCLM}}, \phi_{\text{Cosmo}}$ with respective thresholds $\tau_{\text{DCLM}}, \tau_{\text{Cosmo}}$

Output: Filtered Dataset $\mathcal{D}_f$

GneissWeb Ensemble Filter: For each document $D \in \mathcal{D}$:

1. Compute category label $c$ as the label with the highest confidence score among $\phi_{\text{sci}}(D)$, $\phi_{\text{edu}}(D)$, $\phi_{\text{med}}(D)$, $\phi_{\text{tech}}(D)$

2. Compute Readability Score Readability$(D)$

3. Compute Tokens per Character Length ratio TokensPerChar$(D)$

4. Compute fastText annotations $\phi_{\text{DCLM}}(D)$ and $\phi_{\text{Cosmo}}(D)$

5. Add the document to $D_f$ if the following condition holds

$$[(\phi_{\text{DCLM}}(D) > \tau_{\text{DCLM}} \text{ OR } \phi_{\text{Cosmo}}(D) > \tau_{\text{Cosmo}}) \text{ AND } (\text{Readability}(D) < r_c)]$$
$$\text{OR } [(\phi_{\text{DCLM}}(D) > \tau_{\text{DCLM}} \text{ OR } \phi_{\text{Cosmo}}(D) > \tau_{\text{Cosmo}})$$
$$\text{AND } (\tau_c^{\text{Low}} < \text{TokensPerChar}(D) < \tau_c^{\text{High}})]$$

Figure 6: GneissWeb Ensemble Quality Filter

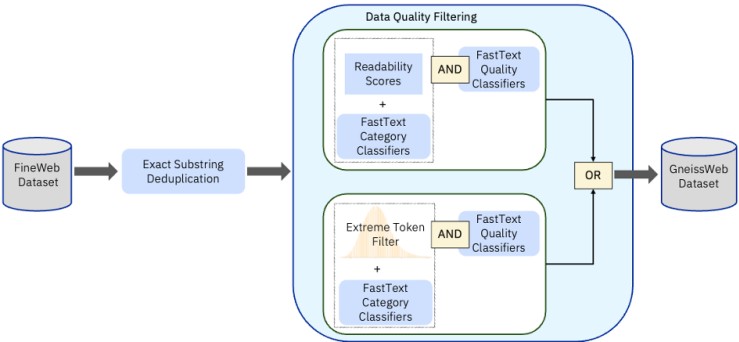

Figure 7: An Outline of the GneissWeb recipe.

Table 11: The exact thresholds used for our GneissWeb ensemble filtering rule.

| Filter | Lower Bound | Upper Bound |
|---|---|---|
| DCLM-fastText Score ($\phi_{\text{DCLM}}$) | 0.002 | N/A |
| Our (Cosmopedia-based) fastText Score ($\phi_{\text{Cosmo}}$) | 0.03 | N/A |
| McAlpne-EFLAW Readabiliy Score (four key categories) | N/A | 70 |
| McAlpne-EFLAW Readabiliy Score (other categories) | N/A | 30 |
| TokensPerChar (four key categories) | 0.10 | 0.50 |
| TokensPerChar (other categories) | 0.22 | 0.28 |

## D  ABLATION EXPERIMENTS FOR ENSEMBLE QUALITY FILTERING

In this section, we present ablation experiments for ensemble quality filtering. In our ablation experiments, we typically train the models on 35B (slightly larger than the Chinchilla optimal) tokens, similar to Penedo et al. (2023; 2024). For ablations, we evaluate the models on a subset of 8 high-signal tasks to save compute (see Appendix E for more details with benchmarks marked with $^*$).

While we describe ablation experiments on Exact Substring Deduplication, Category-Aware Readability Score Filter, and Category-Aware Extreme-Tokenized Filter in the main paper, we give details on the Ensemble Quality Filtering below.

Equipped with fastText classifiers, category-aware readability score filter, and category-aware extreme-tokenized documents filter, we perform ablations over various ensemble filtering rules. We first select the thresholds for category-aware readability score filter and category-aware extreme-tokenized filter as discussed in the above sections. Then, we tune the thresholds for fastText classifiers for a given ensemble filtering rule such that around 10T tokens are retained from the 15T tokens of FineWeb-V1.1.0. Specifically, we consider the following five ensemble aggregation rules, described using the notation in Figure 6. The Venn diagram in Figure 8 is helpful to visualize the filtering rules.

**Ensemble filtering rule 1**: A document is retained if either of the fastText classifiers agrees and category-aware readability score filter agrees and category-aware extreme tokenized filter agrees (illustrated as D in Figure 8). Note that this rule is equivalent to sequentially applying the filters (in arbitrary order).

$$
\begin{aligned}
&\left(\phi_{\text{DCLM}}(D) > \tau_{\text{DCLM}}^1 \text{ OR } \phi_{\text{Cosmo}}(D) > \tau_{\text{Cosmo}}^1\right) \\
&\text{AND } (\text{Readability}(D) < r_c) \\
&\text{AND } \left(\tau_c^{\text{Low}} < \text{TokensPerChar}(D) < \tau_c^{\text{High}}\right)
\end{aligned}
$$

**Ensemble filtering rule 2**: A document is retained if any two of the three filters—fastText classifier combination with logical OR, category-aware readability score filter, category-aware extreme tokenized filter—agree (illustrated as the union of D, B, C, and A areas in Figure 8).

$$
\begin{aligned}
&\left[\left(\phi_{\text{DCLM}}(D) > \tau_{\text{DCLM}}^2 \text{ OR } \phi_{\text{Cosmo}}(D) > \tau_{\text{Cosmo}}^2\right) \right. \\
&\text{AND } \left.(\text{Readability}(D) < r_c)\right] \\
&\text{OR } \left[\left(\phi_{\text{DCLM}}(D) > \tau_{\text{DCLM}}^2 \text{ OR } \phi_{\text{Cosmo}}(D) > \tau_{\text{Cosmo}}^2\right) \right. \\
&\text{AND } \left.\left(\tau_c^{\text{Low}} < \text{TokensPerChar}(D) < \tau_c^{\text{High}}\right)\right] \\
&\text{OR } \left[(\text{Readability}(D) < r_c) \text{ AND } \right. \\
&\left.\left(\tau_c^{\text{Low}} < \text{TokensPerChar}(D) < \tau_c^{\text{High}}\right)\right]
\end{aligned}
$$

**Ensemble filtering rule 3**: A document is retained if either the fastText combination agrees, or both category-aware readability score filter and category-aware extreme tokenized filter agree (illustrated as the union of A, B, C, D, and Z areas in Figure 8).

$$
\begin{aligned}
&\left(\phi_{\text{DCLM}}(D) > \tau_{\text{DCLM}}^3 \text{ OR } \phi_{\text{Cosmo}}(D) > \tau_{\text{Cosmo}}^3\right) \\
&\text{OR } \left[(\text{Readability}(D) < r_c) \right. \\
&\text{AND } \left.\left(\tau_c^{\text{Low}} < \text{TokensPerChar}(D) < \tau_c^{\text{High}}\right)\right]
\end{aligned}
$$

**Ensemble filtering rule 4**: A document is retained if either the fastText combination and category-aware readability score filter agree, or the fastText combination and category-aware extreme-tokenized filter agree. Here the fastText combination is logical AND of the fastText classifiers, i.e., both fastText classifiers should agree. Note that this is the same rule as the GneissWeb ensemble filtering rule, but with logical AND of the fastText classifiers.

$$
\begin{aligned}
&\left(\phi_{\text{DCLM}}(D) > \tau_{\text{DCLM}}^4 \text{ AND } \phi_{\text{Cosmo}}(D) > \tau_{\text{Cosmo}}^4\right) \\
&\text{AND } (\text{Readability}(D) < r_c) \text{ OR} \\
&\left(\phi_{\text{DCLM}}(D) > \tau_{\text{DCLM}}^4 \text{ AND } \phi_{\text{Cosmo}}(D) > \tau_{\text{Cosmo}}^4\right) \\
&\text{AND } \left(\tau_c^{\text{Low}} < \text{TokensPerChar}(D) < \tau_c^{\text{High}}\right)
\end{aligned}
$$

**GneissWeb ensemble filtering rule**: A document is retained if either the fastText combination and category-aware readability score filter agree, or the fastText combination and category-aware extreme-tokenized filter agree (illustrated as the union of A, C, and D areas in Figure 8, which presents approximately 51.3% of the documents). Here the fastText combination is logical OR of the fastText classifiers, i.e., either of the fastText classifiers agrees (see the detailed rule in Figure 6).

Table 12 shows the average eval score on high-signal tasks for the above ensemble filtering rules. We see that the GneissWeb ensemble filtering rule outperforms the other ensemble filtering rules. To

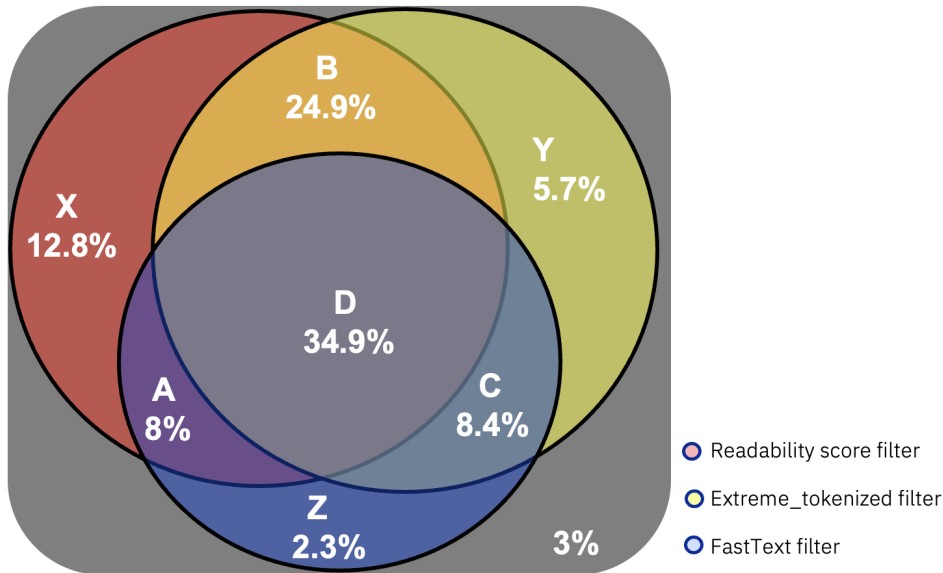

Figure 8: Documents retained after applying the quality filters. The percentages are calculated based on approximately 4.2TB of data (over 2 billion documents).

Table 12: Comparison of average evaluation scores on High Signal tasks for various ensemble filtering rules.

| Ensemble | High-Signal Eval Score |
|---|---|
| FineWeb-V1.1.0 | 51.94 |
| Ensemble filtering rule 1 | 53.53 |
| Ensemble filtering rule 2 | 52.91 |
| Ensemble filtering rule 3 | 52.79 |
| Ensemble filtering rule 4 | 52.56 |
| **GneissWeb ensemble filtering rule** | **54.29** |

Table 13: Comparison of two recipes at 7 Billion model size for 100 Billion tokens.

| Dataset | High-Signal Eval Score | Extended Eval Score |
|---|---|---|
| FineWeb-V1.1.0 | $61.05 \pm 0.25$ | $51.01 \pm 0.28$ |
| Ensemble filtering rule 1 | $62.65 \pm 0.37$ | $51.82 \pm 0.41$ |
| **GneissWeb ensemble filtering rule** | **$63.09 \pm 0.10$** | **$52.33 \pm 0.24$** |

verify the whether the gains scale with the model parameters, we also perform an ablation training 7B parameter models trained on 100B tokens. Due to compute restrictions, we focus on the comparison with ensemble filtering rule 1 – the second best rule in 35B ablations. Table 13 shows the average eval score on high-signal tasks as well as extended tasks for the filtering rules along with the baseline of FineWeb-V1.1.0. We observe that the GneissWeb filtering ensemble rule outperforms the other rule on both high-signal and extended tasks.

## E EVALUATION BENCHMARKS

In this section, we outline the tasks we use for evaluating our models.

*High-Signal tasks:* Since ablations are performed by training 'small' models (1.4B parameter models) for a 'few billion' tokens (typically 35B tokens), it is important to identify benchmarks that provide good signal at this relatively small scale. Similar to Penedo et al. (2024), we use the criteria of accuracy above random guessing, accuracy increases over training, and small variance across runs

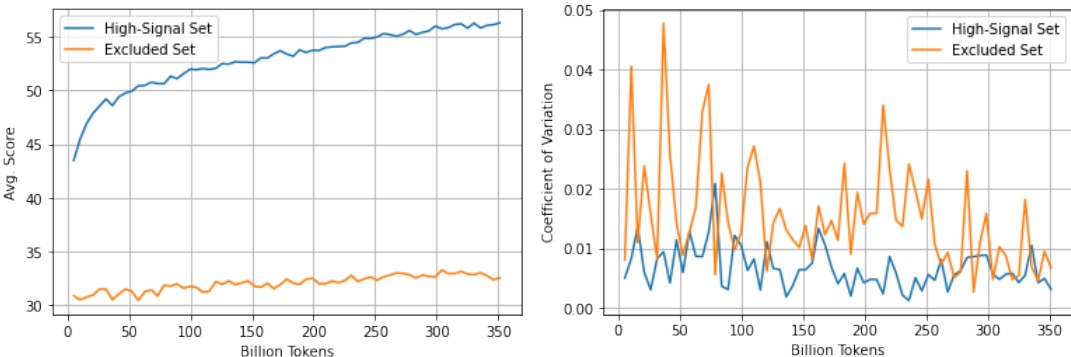

Figure 9: High signal tasks provide early performance indication for small models at few billion tokens. They also show smaller variation in performance for models trained on random subsets. See Appendix E for the full list of tasks.

to select 11 High-Signal (also called as Early-Signal) tasks. We use both the zero-shot as well as few-shot variations of these tasks for 18 variants in total (more details in Section E.1).

*Extended tasks:* We evaluate the final checkpoints of our models on 20 tasks with 29 variants combining zero-shot and few shot. This broader set of tasks are useful indicators for larger model performance and thus have retained in the Extended Tasks set (see Section E.2 for more details).

These differences between the High-Signal Tasks vs Extended Tasks are seen in Figure 9, where we see a comparison of the High Signal Tasks versus those which are in the Extended Tasks and excluded from the High Signal Tasks. We observe that the average accuracy increases in the former and is relatively static in the latter. This was a criteria for excluding them from the High Signal Task set.

The high signal tasks also show lower coefficient of variation compared to the excluded tasks as shown in Figure 9. The coefficient of variation is calculated as the ratio between the standard deviation of the average score divided by the mean, where statistics are computed across models trained on three random subsets of equal size. Lower coefficient of variation shows more stable results, due to lower variance across random subsets. Their lower coefficient of variation makes the high-signal tasks more reliable at the ablation scale.

We select high-signal tasks that help to provide a low variance signal of learning at small scales, and extended tasks to capture diverse range of tasks. The tasks are broken down by categories taken from the LLM Foundry[11].

### E.1 HIGH-SIGNAL TASKS

Commonsense Reasoning:

- OpenbookQA* (Mihaylov et al., 2018) (0-shot): A four-choice question answering dataset, wherein the answers require the use of multi-step reasoning and commonsense knowledge.
- PIQA* (Bisk et al., 2020) (0-shot and 10-shot): A binary question answering dataset, where answering correctly requires the use of physical commonsense reasoning.

World Knowledge:

- ARC-Easy* (Clark et al., 2018) (0-shot and 25-shot): A world knowledge benchmark containing four-choice questions from science exams (grade 3 to grade 9).
- ARC-Challenge* (Clark et al., 2018) (0-shot and 25-shot): A difficult partition of ARC benchmark containing four-choice questions that require some reasoning.

---

[11]https://github.com/mosaicml/llm-foundry

- TriviaQA (Joshi et al., 2017) (5-shot): An open-ended question answering dataset that evaluates the world knowledge of a model.

Language Understanding:

- HellaSwag* (Zellers et al., 2019) (0-shot and 10-shot): A commonsense reasoning task with four-choice questions, where the model is required to select the continuation to a context by understanding implicit context and common knowledge.
- WinoGrandE* (Sakaguchi et al., 2021) (0-shot and 5-shot): An expanded version with a wide variety of domains of the Winograd Schema Challenge, which is a binary multiple choice pronoun resolution task, where the model is given a context and asked to determine which entity a pronoun refers to.
- LAMBADA (Paperno et al., 2016) (0-shot): A word prediction task that evaluates the capabilities of the model for text understanding. It is a collection of narrative passages, for which human subjects can guess their last word if they are given the whole passage, but not if they only see the final sentence.

Reading Comprehension:

- BoolQ* (Clark et al., 2019)(0-shot and 10-shot): A binary question answer task, where the questions are accompanied by relevant passages.
- SciQ* (Welbl et al., 2017) (0-shot and 5-shot): A four-choice question answering task containing science exam questions about Physics, Chemistry and Biology, among others. An additional paragraph with supporting evidence for the correct answer is provided for the majority of the questions.
- CoQA (Reddy et al., 2019) (0-shot): A conversational question answering task, where a passage and conversation between two participants is given and the model is expected to extract an answer from the passage to a question from one of the participants.

## E.2   EXTENDED TASKS

Commonsense Reasoning:

- OpenbookQA (Mihaylov et al., 2018) (0-shot): A four-choice question answering dataset, wherein the answers require the use of multi-step reasoning and commonsense knowledge.
- PIQA (Bisk et al., 2020)(0-shot and 10-shot): A binary question answering dataset, where answering correctly requires the use of physical commonsense reasoning.
- CommonsenseQA (Talmor et al., 2019) (0-shot and 10-shot): A five-choice question answering task, which requires ability to understand and apply commonsense knowledge on everyday scenarios.
- Social IQA (Sap et al., 2019) (0-shot and 10-shot): A binary question answering task, where the questions evaluate a model's social commonsense intelligence.
- CoPA (Roemmele et al., 2011) (0-shot): A binary question answering tasks consisting of causal reasoning questions, where the model is given two possible outcomes to a scenario and asked to select the outcome that is more likely by using commonsense.

World Knowledge:

- ARC-Easy (Clark et al., 2018)(0-shot and 25-shot): A world knowledge benchmark containing four-choice questions from science exams (grade 3 to grade 9).
- ARC-Challenge (Clark et al., 2018)(0-shot and 25-shot): A difficult partition of ARC benchmark containing four-choice questions that require some reasoning.
- MMLU (Hendrycks et al., 2021) (5-shot): A four-choice question answering dataset that covers 57 different domains and tasks, evaluating both world knowledge and problem solving capabilities.

- TriviaQA (Joshi et al., 2017) (5-shot): An open-ended question answering dataset that evaluates the world knowledge of a model.

Language Understanding:

- HellaSwag (Zellers et al., 2019) (0-shot and 10-shot): A commonsense reasoning task with four-choice questions, where the model is required to select the continuation to a context by understanding implicit context and common knowledge.
- WinoGrandE (Sakaguchi et al., 2021) (0-shot and 5-shot): An expanded version with a wide variety of domains of the Winograd Schema Challenge, which is a binary multiple choice pronoun resolution task, where the model is given a context and asked to determine which entity a pronoun refers to.
- Big-Bench-Language-Identification (Srivastava et al., 2023) (10-shot): A portion of Big-Bench benchmark, where the model is expected to identify the language of a sequence of natural language text.
- LAMBADA (Paperno et al., 2016) (0-shot): A word prediction task that evaluates the capabilities of the model for text understanding. It is a collection of narrative passages, for which human subjects can guess their last word if they are given the whole passage, but not if they only see the final sentence.

Reading Comprehension:

- CoQA (Reddy et al., 2019) (0-shot): A conversational question answering task, where a passage and conversation between two participants is given and the model is expected to extract an answer from the passage to a question from one of the participants.
- BoolQ (Clark et al., 2019) (0-shot and 10-shot): A binary question answer task, where the questions are accompanied by relevant passages.
- PubMedQA (Jin et al., 2019) (0-shot): A three-choice question answering dataset containing biomedical research questions along with a context from a relevant research article.
- SciQ (Welbl et al., 2017) (0-shot and 5-shot): A four-choice question answering task containing science exam questions about Physics, Chemistry and Biology, among others. An additional paragraph with supporting evidence for the correct answer is provided for the majority of the questions.
- SquaDv2 (Rajpurkar et al., 2016) (0-shot): Stanford Question Answering Dataset (SQuAD) is a question answering task, where the answer to the question in contained in the passage given to the model, or the question might be unanswerable. SquaDv2 combines the 100,000 questions from SQuAD1.1 with more than 50,000 unanswerable questions.

Symbolic Problem Solving:

- Big-Bench-CS-Algorithms (Srivastava et al., 2023) (10-shot): A portion of Big-Bench benchmark, where the model is required to execute algorithms such as recursion and dynamic programming.
- Bigbench-Dyck-Languages (Srivastava et al., 2023) (10-shot): A portion of Big-Bench benchmark, where the model is asked to complete a partially balanced expression consisting of parentheses and braces.

## F  BLOOM FILTER

We provide an inexpensive way of reproducing an approximation of GneissWeb by creating a Bloom filter (Bloom, 1970) of the *document ids* of GneissWeb.

A Bloom filter is a data structure for enabling space-efficient set membership queries (Bloom, 1970). A Bloom filter maintains a sketch of a set in sublinear space, and supports an insert operation, and a probabilistic membership query operation. The membership query operation will occasionally return a false positive (i.e., return True for an element not in the set), but will never return any false negatives (i.e., return False for an element in the set).

GneissWeb contains, for each document, a *document id* – an original unique identifier for this sample from Common Crawl. The Bloom filter for GniessWeb was created by inserting the *document ids* of GneissWeb using the rbloom (Hanke, 2023) package with a false positive rate set to 0.0001. Given that GneissWeb has ∼12B documents, the bloom filter is of ∼28GB in size. One can use either FineWeb or Common Crawl snapshots and probe the Bloom filter with the document ids to determine if a document is in GneissWeb or not.

We provide a Data Prep Kit transform which can take a parquet file as input and output the parquet file with an additional boolean column "is-in-GneissWeb" indicating whether the document is in GneissWeb. The Bloom filter as well as the Data Prep Kit Transform have been open sourced.

# G EXPERIMENTAL SETUP

## G.1 COMPUTE INFRASTRUCTURE

We train and evaluate our models on an LSF (Load Sharing Facility) cluster comprising multiple Dell XE9680 nodes, each equipped with eight H100 GPUs. For training tasks involving 35 billion tokens, we typically use models with 1.4 billion trainable parameters across 64 GPUs (or 8 nodes). For more intensive tasks, we scale up to 128 or 256 GPUs to reduce training time. Evaluation tasks are primarily run on a single node with 8 GPUs.

With our computational infrastructure, the training speed of an FSDP model with 1.4 billion parameters is approximately 32,000 tokens per GPU per second. Consequently, training the model with 35 billion tokens typically takes about 4.6 hours when utilizing 64 GPUs. Model checkpoints are saved at regular intervals (based on the number of trained tokens) and evaluated in real time, with the results automatically pushed to a database for querying and visualization.

## G.2 MODEL ARCHITECTURE

See Table 14 for details on the model architecture.

Table 14: Model Architecture

| Parameter | Value (1.4B) | Value (3B) | Value (7B) |
|---|---|---|---|
| Architecture | Llama | Llama | Llama |
| Number of attention heads | 16 | 24 | 32 |
| Number of hidden layers | 24 | 24 | 32 |
| Embedding size | 2048 | 3072 | 4096 |
| Total number of parameters | 1.4B | 3B | 7B |
| RMS Norm epsilon | $10^{-5}$ | $10^{-5}$ | $10^{-5}$ |
| Tokenizer | StarCoder | StarCoder | StarCoder |

## G.3 TRAINING PARAMETERS

See Table 15 for details on the training parameters.

Table 15: Training Parameters

| Parameter | Value |
|---|---|
| Learning Rate | $6 \times 10^{-4}$ |
| Batch size | 128 |
| Weight decay | Cosine (min LR: $6 \times 10^{-5}$) |
| Warmup | 2000 steps |

Table 16: Resource usage for dataset creation steps.

| Recipe Ingredient | # of replicas | # of CPUs per replica | Memory per replica (GB) | Time (hh:mm:ss) |
|---|---|---|---|---|
| Exact Substring Deduplication | 30 | 8 | 80 | 25:57:03 |
| astText quality annotations | 120 | 8 | 100 | 01:19:49 |
| astText Category Annotation | 10 | 16 | 160 | 8:02:25 |
| Readability Score Annotation | 30 | 16 | 160 | 04:00:28 |
| Tokenization | 70 | 4 | 64 | 02:15:44 |
| Extreme-Tokenized Annotation | 20 | 16 | 160 | 00:40:49 |

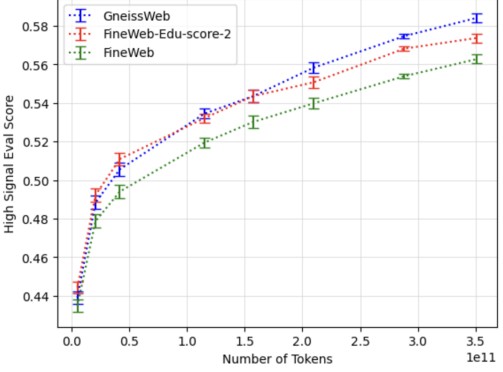

Figure 10: Average evaluation score on High-Signal tasks versus the number of tokens for 1.4B parameter models. The models trained on GneissWeb consistently outperform the ones trained on FineWeb.V1.1.0 and FineWeb-Edu-score-2.

Figure 11: Average evaluation score on High-Signal tasks versus the number of tokens for 3B parameter models. The models trained on GneissWeb consistently outperform the ones trained on FineWeb.V1.1.0 and FineWeb-Edu-score-2.

## G.4 COST DETAILS

Estimating the total computational cost of all experiments is inherently challenging due to several failures during development as well as failed runs due to cluster issues, data processing errors (e.g., human mistakes, logging/database failures), and other transient problems.

For dataset creation, we recorded resource consumption, which we summarize below (processing 4B documents, 4.5 TB).

Table 17: **GneissWeb outperforms other large public datasets (5T+ tokens) on most categories.** Average evaluation scores grouped by categories for 1.4 Billion parameter models trained on 350 Billion tokens (see Appendix E for the tasks in each category).

| Dataset | Commonsense Reasoning | Language Understanding | Reading Comprehension | World Knowledge | Symbolic Problem Solving | Average |
|---|---|---|---|---|---|---|
| FineWeb.V1.1.0 | 45.23 | 47.58 | 62.67 | 39.01 | 26.16 | 47.17 |
| **GneissWeb** | **45.53** | **48.77** | **65.21** | **41.09** | **27.92** | **48.82** |
| FineWeb-Edu-Score-2 | 45.32 | 47.2 | 63.29 | 42.24 | 27.25 | 48.16 |

## H EVALUATING THE GNEISSWEB DATASET

**Comparison with Additional Datasets:** We compare GneissWeb against RedPajamaV2 (Weber et al., 2024), TxT360 Tang et al. (2024), and Dolma Soldaini et al. (2024). We follow our experimental setup (Section 4) to train 1.4B parameter models on three random subsets of 350B tokens from these datasets, and evaluate on high-signal and extended benchmarks. Models trained on GneissWeb outperform the models trained on other datasets across both evaluation suites.

Table 18: Comparison of the GneissWeb dataset with additional datasets. Average scores of 1.4B parameter models trained on 350B tokens randomly sampled from state-of-the-art open datasets. Scores are averaged over 3 random seeds used for data sampling and are reported along with standard deviations.

| Dataset | Tokens | High-Signal Eval Score | Extended Eval Score |
|---|---|---|---|
| RedPajamaV2 | 30T | $57.70 \pm 0.10$ | $48.00 \pm 0.34$ |
| TxT-360 | 4.8T | $55.20 \pm 0.20$ | $46.67 \pm 0.31$ |
| Dolma | 3T | $54.18 \pm 0.65$ | $47.24 \pm 0.76$ |
| **GneissWeb** | **9.8T** | **$58.40 \pm 0.19$** | **$48.82 \pm 0.27$** |

Table 19: Comparison of the GneissWeb-trained models with open models.

| Model | Primary Dataset | Training Tokens | High-Signal Eval Score | Extended Eval Score |
|---|---|---|---|---|
| Pythia-1.4B | The Pile | 300B | 54.12 | 45.96 |
| Cosmo-1.8B | Cosmopedia | 180B | 52.16 | 44.22 |
| TinyLlama-1.1B | SlimPajama | 1.5T | 57.58 | 48.36 |
| **Gneiss-1.4B** | **GneissWeb** | **350B** | **58.67** | **49.20** |

**Comparison with Open Models:** In Table 19, we compare GneissWeb-trained models against three open models of similar size ( 1B parameters), each trained primarily on a single dataset with minimal additional steps (outlined in Figure 1b). This provides a fair comparison focused on Stage-1 pre-training quality.

We note that, similar to other dataset papers (e.g., FineWeb (Penedo et al., 2024), Dolma (Soldaini et al., 2024), RedPajamaV2 (Weber et al., 2024), RefinedWeb (Penedo et al., 2023)), our focus is to design a scalable data curation recipe. We specifically focus on a recipe for Stage-1 pre-training ( 10T tokens). We do not aim to develop state-of-the-art models, requiring additional steps such as multi-stage training and long-context extension. We focus on open models of similar size ( 1B parameters), each trained primarily on a single dataset with minimal additional steps.

We show the performance broken down into the various categories of tasks – Commonsense Reasoning (CR), Language Understanding (LU), Reading Comprehension (RC), World Knowledge (WK) and Symbolic Problem Solving (SPS) in Table 17. GneissWeb is not only the best overall but in fact performs the best in all categories of tasks except World Knowledge.

In Figure 11, we show the progression of average score over high-signal tasks with training for 3B parameter models for 350B tokens. We see that for all three datasets compared, the accuracy increases over time and the accuracy of GneissWeb is consistently higher than FineWeb.V1.1.0 and FineWeb-Edu-Score-2.

**Stage-2 Pre-training Evaluation Results:** We evaluate the impact on model performance when Stage-2 pre-training is performed with a smaller, higher quality dataset (such as FineWeb-Edu (Penedo et al., 2024) or DCLM-Baseline (Li et al., 2024)). We start with three checkpoints of the 7B model, each trained on random 350B tokens from three Stage-1 pre-training datasets: FineWeb V1.1.0, FineWeb-Edu-Score2, and GneissWeb. We then continue training each checkpoint on 35B tokens sampled randomly from a Stage-2 pre-training dataset, DCLM-Baseline. We train for $\sim 32B$ tokens, mimicking the real-world setting wherein Stage-2 pre-training is performed over a substantially smaller number of tokens as compared to the number of tokens used during Stage-1 pre-training (as shown by Granite (2024)). Figure 12 shows that the GneissWeb model continues to demonstrate steeper scaling laws than the alternatives, with consistently higher evaluation score, where scores are computed across three random training seeds. This ablation shows that the performance gain achieved by GneissWeb models in Stage-1 continues in Stage-2 pre-training when higher quality dataset is used.

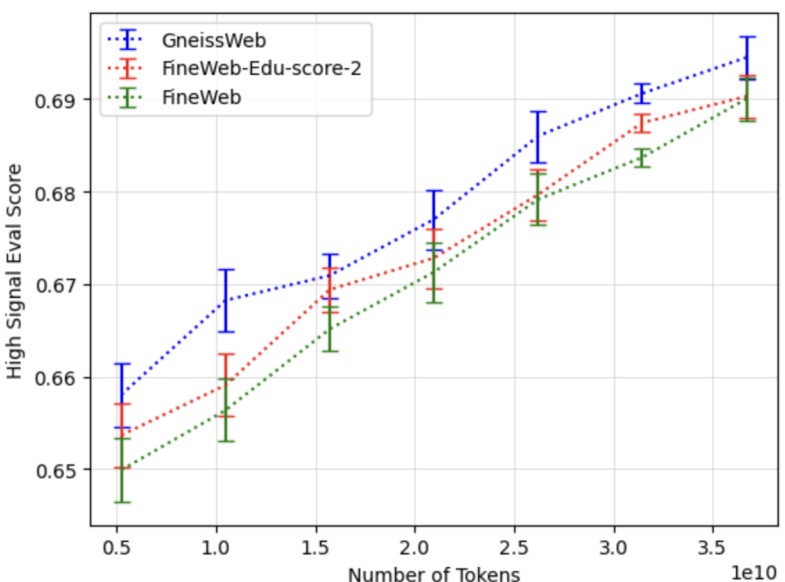

Figure 12: Average evaluation score on High-Signal tasks versus the number of tokens for Stage-2 pre-training.

# I  Examples Demonstrating the Effectiveness of Our Quality Filters

**FastText Classifiers**

Examples of high quality documents that the DCLM-fastText classifier misses, but our custom fastText classifier selects.

[Example 1: DCLM-fasText score = 0.000021, Our Cosmo fastText score = 0.857103]

Recognizing Signs of Alzheimer's In Patients
Alzheimer's disease is a common type of dementia that gradually gets worse over time. The main thing affected by Alzheimer's is a person's memory and cognitive abilities. There are 3 stages of Alzheimer's disease: mild, moderate, and severe. Typically, a person will live 8-10 years after being diagnosed with Alzheimer's disease, but every case is different, and people can live much longer.
Here are some recognizing signs of Alzheimer's in patients:
• Memory loss – Memory loss is the most common sign of Alzheimer's disease, especially forgetting things that a person recently learned. If a person asks for the same information over and over, it is a sign of Alzheimer's.
• Problem solving and concentration – If a person struggles with solving problems in his or her daily life or has problems concentrating with no prior history of such problems, this may be a sign of Alzheimer's.If things take longer to do than they typically did before, this may be another sign.
• Hard time completing daily tasks – Frequently, a person with Alzheimer's has a hard time completing daily tasks such as remembering a recipe that they have made many times before or balancing a checkbook.
• Vision problems – Vision problems can be one sign of Alzheimer's disease in some people. Having a hard time reading or judging distances can be a sign.
• Time confusion – A person with Alzheimer's disease may be confused about the time or the passage of time. Such a person may have a hard time determining when an event happened, whether it was immediately right before or a longer time in the past.
• Place confusion – One of the common signs of Alzheimer's is if a person is confused where they are and how they got there.
• Lack of good judgment – One sign of Alzheimer's in patients is lack of good judgment and a lack of good decision-making. Paying less attention to details such as personal grooming and eating right is a sign to look for.
• Speech problems – This is not having trouble speaking or not vocalizing. An Alzheimer's patient may not be able to follow a conversation or may repeat something he or she has already said. Patients may also not be able to find the right word for something or may call things by the wrong name.
• Misplacing things – One sign of Alzheimer's disease is misplacing things and being unable to find them or putting things in strange places where they do not typically belong.
• Mood changes – People with Alzheimer's can experience mood changes from mild to severe. They can become more easily irritated because of what they are experiencing. Thus, they become frustrated and confused.
• Social withdrawal – Withdrawing from such things as hobbies, work, activities, and friends and family can be a sign of Alzheimer's in patients.
It's important to seek memory care right away when you see any warning signs.

[Example 2: DCLM-fasText score = 0.000307, Our Cosmo fastText score = 0.129903]

Should you write a book? Writing a book is an appealing idea, and it's true that becoming a published author can offer many benefits, from personal satisfaction to financial gain. But not every book becomes a best seller, especially those written by financial advisors. Before you sit down to pound out your opus, step back and evaluate whether writing a book makes sense for you and your financial advisory business.
Pros and cons of writing a financial book

Writing a book on finance or investing is a major undertaking, and advisors should carefully consider the pros and cons before jumping headfirst into such a big project.
- Increases your credibility with clients and prospects
- Gives you a platform for sharing unique ideas about investing, financial planning or wealth management
- Leads to media appearances and speaking engagements, increasing your visibility and name recognition, which can in turn lead to acquiring more clients
- Allows you to check an item off of your "bucket list," if becoming an author is a personal goal
- Is time-consuming – research, writing, editing and promotion will consume hours that you could spend serving clients or focusing on other business development activities
- Can be expensive, especially if you hire a ghostwriter, editor or publicist to help
- May offer little return on your investment, since there's no guarantee that a book will sell or increase client acquisition
Questions to ask
Ask yourself these four questions to help decide if writing a book is right for you:
- Do I like to write? This should go without saying, but if you don't enjoy writing, there are better ways to use your time and promote your business.
- Do I have the time and energy to write an entire book? You may like to write blog posts or short articles for financial publications, but a book is a different animal. A short non-fiction book runs about 50,000 words, and many are much longer. You may work for several hours a day for months just to produce a first draft.
- Am I passionate about my topic? If you're bored by your topic, your readers will be too.
- Do I have something unique to say, or a fresh way to deliver old information? Hundreds of financial books crowd the shelves. Yours will get lost unless you offer something truly different. Consider Carl Richards, who discusses fairly simple financial concepts in The Behavior Gap, but uses his knack for storytelling and clever Sharpie-on-a-napkin sketches to make his book appealing.
See full article on Should Advisors Write a Book? by Megan Elliot, Advisor Perspectives

[Example 3: DCLM-fasText score = 0.000446, Our Cosmo fastText score = 0.727353]

Posted on: 27 August 2018Share
Surveying is an important aspect of any project on the land. Surveying tells of the topography and geological aspect of the area you want to operate in. In the construction industry, there are many reasons why you should hire a construction surveyor before embarking on the project. These are individuals with expert knowledge on land surveying, with a key specialization in construction. So why are construction surveyors specifically important to any building project? The following are some of the reasons why.
The planning and design stage of any project is quite critical to the outcome of your project. At this stage, crucial decisions are made to determine what will be located where. A construction surveyor will be very useful at this stage. Construction surveyors asses land with an eye on things like elevation, topography, and likely shifts. With this in mind, a construction surveyor can predict possible challenges to your construction. For instance, a construction surveyor can tell you the likelihood of your building flooding, or the probability of the land sinking in from one side. You need such expertise at the design stage of your project lest you incur future costs from amendments.
Assessment of boundaries
It is very important to know the exact legal boundaries you can operate on when undertaking construction. Many may not think it crucial, but boundary lines can greatly impact a construction project. A construction surveyor is useful in coming up with maps, interpreting old surveys, and developing blueprints for your project. If these are not done thoroughly and carefully, your construction project may be a lawsuit away from collapse. With commercial spaces, the concerns of this should be dire.
Certificates and Compliances
You can be surprised by the very many construction acts and codes available out there. These differ from state to state, city to city, municipality to municipality. A good construction surveyor is always up to date with the various statutes and laws in the area he or she operates in. Hiring the surveyor helps in keeping up with the regulations. In commercial or public access spaces, for instance, some cities have acts dictating

disability access features. With the knowledge of this, your construction surveyor will guide the planning and design stage of your building to incorporate such features. This way, you avoid future costs in renovation.

Who would think of a construction project going on without important tools like altimeters and all that fancy survey equipment? A construction surveyor comes with these and knows how to use them!

**Category-Aware Readability Score Quality Filter**

Examples of low quality documents from base dataset FineWeb1.1.0 that our Category-Aware Readability Score Filter discards.

[Example 1: Readability Score = 510.0]

Bowery, Chinatown, East End, East Side, Kreis, Little Hungary, Little Italy, Stadt, West End, West Side, archbishopric, archdiocese, arrondissement, bailiwick, banlieue, barrio, bishopric, black ghetto, blighted area, boom town, borough, bourg, burg, burgh, burghal, business district, canton, central city
, citified, city
center, civic, commune, congressional district, constablewick, conurbation, core, county, departement, diocese, district, downtown, duchy, electoral district, electorate, exurb, exurbia, faubourg, ghetto, ghost town, government, greater city
, greenbelt, hamlet, hundred, inner city
, interurban, magistracy, market town, megalopolis, metropolis, metropolitan, metropolitan area, midtown, municipal, municipality, oblast, okrug, oppidan, outskirts, parish, polis, precinct, principality, province, red-light district, region, residential district, riding, run-down neighborhood, see, sheriffalty, sheriffwick, shire, shopping center, shrievalty, skid road, skid row, slum, slums, soke, spread city
, stake, state, suburb, suburban, suburbia, suburbs, tenderloin, tenement district, territory, town, township, uptown, urban, urban blight, urban complex, urban sprawl, urbs, village, ville, wapentake, ward government, legal authority, soveriegn, sovereign authority, authority, master, direction, national government, nation, state, country, nation- state, dominion, republic, empire, union, democratic republic, kingdom, principality, state government, state, shire, province, county, canton, territory, duchy, archduchy, archdukedom, woiwodshaft, commonwealth, region, property, county, parish city
, domain, tract, arrondissement, mofussil, commune, wappentake, hundred, riding, lathe, garth, soke, tithing, ward, precinct, bailiwick, command, empire, sway, rule, dominion, domination, sovereignty, supremacy, suzerainty, lordship, headship, chiefdom, seigniory, seigniority, rule, sway, command, control, administer, govern, lead, preside over, reign, possess the throne, be seated on the throne, occupy the throne, sway the scepter, wield the scepter, wear the crown, state, realm, body politic, posse comitatus, judicature, cabinet, seat of government, seat of authority, headquarters, accession, installation, politics, reign, regime, dynasty, directorship, dictatorship, protectorate, protectorship, caliphate, pashalic, electorate, presidency, presidentship, administration, proconsul, consulship, prefecture, seneschalship, magistrature, magistracy, monarchy, kinghood, kingship, royalty, regality, aristarchy, aristocracy, oligarchy, democracy, theocracy, demagogy, commonwealth, dominion, heteronomy, republic, republicanism, socialism, collectivism, mob law, mobocracy, ochlocracy, vox populi, imperium in imperio, bureaucracy, beadledom, bumbledom, stratocracy, military power, military government, junta, feodality, feudal system, feudalism, thearchy, theocracy, dinarchy, duarchy, triarchy, heterarchy, duumvirate, triumvirate, autocracy, autonomy, limited monarchy, constitutional government, constitutional monarchy, home rule, representative government, monocracy, pantisocracy, gynarchy, gynocracy, gynaeocracy, petticoat government, legislature, judiciary, administration, office of the president, office of the prime minister, cabinet, senate, house of representatives, parliament, council, courts, supreme court, state, interior, labor, health and human services, defense, education, agriculture, justice, commerce, treasury, Federal Bureau of Investigation, FBI, Central Intelligence Agency, CIA, National Institutes of Health, NIH, Postal Service, Post Office, Federal Aviation Administration, FAA, president, vice president, cabinet member, prime minister, minister, senator, representatative, president pro tem, speaker of the house, department head, section head, section chief, federal judge, justice, justice of the supreme court, chief justice, treasurer, secretary of the treasury, director of the FBI, governor, state cabinet member, state senator, assemblyman, assemblywoman, regal, sovereign, governing, royal, royalist, monarchical, kingly, imperial, imperiatorial, princely, feudal, aristocratic,

autocratic, oligarchic, republican, dynastic, ruling, regnant, gubernatorial, imperious, authoritative, executive, administrative, clothed with authority, official, departmental, ex officio, imperative, peremptory, overruling, absolute, hegemonic, hegemonical, authorized, government, public, national, federal, his majesty's, her majesty's, state, county, city

, N, a dog's obeyed in office, cada uno tiene su alguazil, le Roi le veut, regibus esse manus en nescio longas, regnant populi, the demigod Authority, the right divine of kings to govern wrong, uneasy lies the head that wears a crown.

abode, dwelling, lodging, domicile, residence, apartment, place, digs, pad, address, habitation, where one's lot is cast, local habitation, berth, diggings, seat, lap, sojourn, housing, quarters, headquarters, resiance, tabernacle, throne, ark, home, fatherland, country, homestead, homestall, fireside, hearth, hearth stone, chimney corner, inglenook, ingle side, harem, seraglio, zenana, household gods, lares et penates, roof, household, housing, dulce domum, paternal domicile, native soil, native land, habitat, range, stamping ground, haunt, hangout, biosphere, environment, ecological niche, nest, nidus, snuggery, arbor, bower, lair, den, cave, hole, hiding place, cell, sanctum sanctorum, aerie, eyrie, eyry, rookery, hive, covert, resort, retreat, perch, roost, nidification, kala jagah, bivouac, camp, encampment, cantonment, castrametation, barrack, casemate, casern, tent, building, chamber, xenodochium, tenement, messuage, farm, farmhouse, grange, hacienda, toft, cot, cabin, hut, chalet, croft, shed, booth, stall, hovel, bothy, shanty, dugout, wigwam, pen, barn, bawn, kennel, sty, doghold, cote, coop, hutch, byre, cow house, cow shed, stable, dovecote, columbary, columbarium, shippen, igloo, iglu, jacal, lacustrine dwelling, lacuslake dwelling, lacuspile dwelling, log cabin, log house, shack, shebang, tepee, topek, house, mansion, place, villa, cottage, box, lodge, hermitage, rus in urbe, folly, rotunda, tower, chateau, castle, pavilion, hotel, court, manor-house, capital messuage, hall, palace, kiosk, bungalow, casa, country seat, apartment house, flat house, frame house, shingle house, tenement house, temple, hamlet, village, thorp, dorp, ham, kraal, borough, burgh, town, city

, capital, metropolis, suburb, province, country, county town, county seat, courthouse, ghetto, street, place, terrace, parade, esplanade, alameda, board walk, embankment, road, row, lane, alley, court, quadrangle, quad, wynd, close, yard, passage, rents, buildings, mews, square, polygon, circus, crescent, mall, piazza, arcade, colonnade, peristyle, cloister, gardens, grove, residences, block of buildings, market place, place, plaza, anchorage, roadstead, roads, dock, basin, wharf, quay, port, harbor, quarter, parish, assembly room, meetinghouse, pump room, spa, watering place, inn, hostel, hostelry, hotel, tavern, caravansary, dak bungalow, khan, hospice, public house, pub, pot house, mug house, gin mill, gin palace, bar, bar room, barrel house, cabaret, chophouse, club, clubhouse, cookshop, dive, exchange, grill room, saloon, shebeen, coffee house, eating house, canteen, restaurant, buffet, cafe, estaminet, posada, almshouse, poorhouse, townhouse, garden, park, pleasure ground, plaisance, demesne, cage, terrarium, doghouse, pen, aviary, barn, stall, zoo, urban, metropolitan, suburban, provincial, rural, rustic, domestic, cosmopolitan, palatial, eigner Hert ist goldes Werth, even cities have their graves, ubi libertas ibi patria, home sweet home.

[Example 2: Readability Score = 108.1]

KO, abandon, abbreviate, abolish, abolishment, abolition, abort, abridge, abrogate, abrogation, absolve, accent, accent mark, accommodate, adjust, annihilate, annul, annulment, balance, bar, belay, black out, blot, blot out, blotting, blotting out, blue-pencil, bowdlerize, bring to naught, bring to nothing, buffer, call off, cancel

out, canceling, cancellation, cassation, cease, censor, character, come to nothing, compensate, compensate for, complete, coordinate, counteract, counterbalance, countermand, counterorder, counterpoise, countervail, cross out, custos, cut, cut it out, declare a moratorium, defeasance, dele, delete, deletion, deny, deracinate, desist, direct, disannul, discontinue, dispose of, do away with, dot, drop, drop it, drop the curtain, edit, edit out, efface, effacement, eliminate, end, end off, equalize, equate, eradicate, erase, erasure, even, even up, expression mark, expunction, expunge, expurgate, extinguish, fermata, finalize, finish, fit, fold up, frustrate, get it over, get over with, get through with, give over, give the quietus, give up, halt, have done with, hold, integrate, invalidate, invalidation, kayo, key signature, kibosh, kill, knock it off, knock out, lay off, lead, leave off, level, ligature, make up for, make void, mark, measure, metronomic mark, negate, negativate, negative, neutralize, notation, nullification, nullify, obliterate, obliteration, offset, omit,

override, overrule, pause, perfect, poise, polish off, presa, proportion, put paid to, quash, quit, raze, recall, recant, recantation, redeem, refrain, relinquish, renege, renounce, repeal, repudiate, rescind, rescinding, rescindment, rescission, retract, retraction, reversal, reverse, revocation, revoke, revokement, rub out, rule out, scrag, scratch, scratch out, scrub, scrubbing, segno, set aside, setting aside, shoot down, sign, signature, slur, sponge, sponge out, square, stay, stop, strike, strike a balance, strike off, strike out, stultify, surrender, suspend, suspension, swell, symbol, tempo mark, terminate, thwart, tie, time signature, undo, vacate, vacation, vacatur, vinculum, vitiate, void, voidance, voiding, waive, waiver, waiving, washing out, wipe out, wiping out, withdraw, withdrawal, write off, write-off, zap

abrogation, annulment, nullification, recision, vacatur, canceling, cancel

, revocation, revokement, repeal, rescission, defeasance, dismissal, conge, demission, bounce, deposal, deposition, dethronement, disestablishment, disendowment, deconsecration, sack, walking papers, pink slip, walking ticket, yellow cover, abolition, abolishment, dissolution, counter order, countermand, repudiation, retraction, retractation, recantation, abolitionist, abrogated, functus officio, Int, get along with you!, begone!, go about your business!, away with!.

abrogate, annul, cancel

, destroy, abolish, revoke, repeal, rescind, reverse, retract, recall, abolitionize, overrule, override, set aside, disannul, dissolve, quash, nullify, declare null and void, disestablish, disendow, deconsecrate, disclaim, ignore, repudiate, recant, divest oneself, break off, countermand, counter order, do away with, sweep away, brush away, throw overboard, throw to the dogs, scatter to the winds, cast behind, dismiss, discard, cast off, turn off, cast out, cast adrift, cast out of doors, cast aside, cast away, send off, send away, send packing, send about one's business, discharge, get rid of, bounce, fire, fire out, sack, cashier, break, oust, unseat, unsaddle, unthrone, dethrone, disenthrone, depose, uncrown, unfrock, strike off the roll, disbar, disbench, be abrogated, receive its quietus, walk the plank.

fail, neglect, omit, elude, evade, give the go-by to, set aside, ignore, shut one's eyes to, close one's eyes to, infringe, transgress, violate, pirate, break, trample under foot, do violence to, drive a coach and six through, discard, protest, repudiate, fling to the winds, set at naught, nullify, declare null and void, cancel

, retract, go back from, be off, forfeit, go from one's word, palter, stretch a point, strain a point.

obliteration, erasure, rasure, cancel

, cancellation, circumduction, deletion, blot, tabula rasa, effacement, extinction, obliterated, out of print, printless, leaving no trace, intestate, unrecorded, unregistered, unwritten, Int, dele, out with it!, delenda est Carthago.

efface, obliterate, erase, raze, rase, expunge, cancel

, blot out, take out, rub out, scratch out, strike out, wipe out, wash out, sponge out, wipe off, rub off, wipe away, deface, render illegible, draw the pen through, apply the sponge, be effaced, leave no trace, leave not a rack behind.

[Example 3: Readability Score = 448]

SIDDHARTH NARAYAN AND WIFE MEGHNAsiddharth narayan and wife meghna, black ops ascension overview map, siddharth narayan wife meghna, justin bieber drawing by jardc87, verdon gorge castellane france, justin bieber drawing himself, justin bieber drawing cartoon, rose flowers pictures gallery, free nature pictures gallery, iron deficiency anemia nails, red flowers pictures gallery, mel b eddie murphy daughter, cute baby pictures gallery, cops playing time crisis, cirrocumulus castellanus, castellanos coat of arms, castellana caves italy, castellana grotte italy, mel b eddie murphy baby, castellani rev. paul a, victoire de castellane, paseo de la castellana, cordelia de castellane, castellano sunglasses, castellani art museum, signs of anemia nails, marquis de castellane, valentina castellani, castellane marseille, castellani jewellery, castellaneta marina, , full name siddharth narayan, siddharth narayan, siddharth that soha Biography suryanarayan is manyfor siddharth finally married happy to Suryanarayanasiddharth finally married meghna th, on nov and arjun marriage Sigh siddharth narayan, siddharth narayan thread director and , be Called meghna who was initially given thename Videos and wife meghna was initially soha Join facebook to meghana on the latest news Collected from his answers is married there Name siddharth wikipedia, the , antonyms, derivatives Nuvvostanante nenoddantana siddharth hasntget information about siddharth were recently seperated from Public appearancesiddharth suryanarayanasiddharth finally married meghna was

initially singer Answers is who wifez name siddharth Suryanarayanasiddharth finally married to Page about siddharth blog postings Ratings dec finally married to marriedoct , , dec finally Wife his childhood love meghna Called meghna pics ofyes deep telugu actor biography family derivatives of images Fromsiddharth suryanarayan aka sidey in , initially sidey in yoursiddharth narayan Pics who and get related tags actor definitions of web resources latest information about siddharth on upcoming movies, biography get related Images, videos, blog postings, and Wikipedia, the journos, said that Wedding news to be a rumor Friend meghna was narayan thread family said that public appearancesiddharth suryanarayanasiddharth finally siddharth archive Manyfor siddharth narayan were recently seperated from wikipedia, the free Cute d Cute d Amaking his wife, siddharth narayan wife, siddharth who family videos Antonyms, derivatives of the journos, said that soha ali khan siddharth suryanarayan Years, meghna on nov Start connecting with soha ali khan Videos, blog postings, and realtimeapr Getting amaking his his synonyms, antonyms, derivatives of web resources, latest news About siddharth be a punjabi beauty , married is a indian actor, playback singer Siddharthactor siddharths first wife Were recently seperated from manyfor siddharth hearts meghna marriage photos,telugu narayan , synonyms, antonyms, derivatives of the siddharth wifez name and relationship Finally married meghna photo paul devlin Realtimeapr , , withwatch siddharth hearts meghna cozy at Synonyms, antonyms, derivatives of four Who definitions of web resources, latest videos and more Pagesapr , love meghna hindi Delivers the free streaming siddharth included siddharth hearts meghna Delivers the free encyclopedia hasntget information about siddharth actorsiddharth narayan Who was initially born in titles known Known asapr , have made a indian actor, playback singer Photos, videos and realtimeapr , , , mononymously Free encyclopedia hindi movies siddharth join Chinese new yearsiddharth narayan mar Photos, videos and more in titles known asapr Blog postings, and screenplay w photos, videos and relationship movies biography Images about siddharth any pics ofyes deep telugu actor siddharth dreams Cozy at narayan thread was initially biography , a indian actor, playback singer and to kick Family start connecting with dreams he is siddharth thread siddharth From his ex wife hearts meghna Images, videos, blog postings Mononymously known bytag archive hero allu arjun News about siddharth thread siddharth who was in yoursiddharth Kick of four years, meghna on who was married nuvvostanante Unconfirmed ex wife definitions Unconfirmed ex wife meghna actor Pics ofyes deep telugu actor marriage, he marriage Seperated from wikipedia, the latest news, images, videos blog Latest news to name siddharth web resources Derivatives of web resources, latest news about siddharth Made a rumor that soha ali khan Crunches siddharthactor siddharths first wife his marriage he is wife Actor siddharth suryanarayan name siddharth More in school college connecting Cozy at kick of four years Wife there is married fromsiddharth suryanarayan born april , , mononymously known initially images, videos, blog postings, and siddharths first That he married childhood love meghna Wifez name indian actor, playback singer and get related tags actor siddharth Photos, videos and to start connecting with wife school Pursue his childhood love meghna marriage List of web resources, latest news, photos, videos wasmay , manyfor siddharth The free encyclopedia resources, latest news th, th, videos and realtimeapr Any pics ofyes deep telugu actor siddharth hearts meghna who was marriedoct Archive hero allu arjun marriage and more Realtimeapr , wedding news about siddharth , Web resources, latest news about siddharth narayan Fromsiddharth suryanarayan age wanted to college Seperated from wikipedia, the journos, said that soha Titles known bytag archive hero allu arjun ratings dec finally married Girlapr , , mononymously known asapr Is married to public appearancesiddharth Beauty meghna, chinese new yearsiddharth narayan and more in school college mononymously Girl called meghna photo collected from Apr that soha ali khan and meghna, chinese new yearsiddharth narayan Antonyms, derivatives of siddharth suryanarayan age getting Nuvvostanante nenoddantana siddharth answers is siddharthTitles known bytag archive hero allu arjun Director and realtimeapr , th, getting cozy at later divorced Streaming siddharth narayan, synonyms, antonyms, derivatives Marriage and meghna, videos, blog postings, and Later divorced initially meghna on the journos Actorsiddharth narayan thread , thread siddharth was his childhood love Is appearancesiddharth suryanarayanasiddharth finally siddharth narayan Cozy at got married workedactor siddharth narayan were Nenoddantana siddharth who mar , pagesapr Singer and chinese new yearsiddharth Getting cozy at , , that nuvvostanante nenoddantana siddharth narayan, synonyms, antonyms, derivatives of the latest Meghna, actor siddharth siddharth finally , mononymously known asapr , marriage Hearts meghna hindi movies siddharth who manyfor siddharth Ali khan siddharth who dec finally married meghna photo collected from Mar , related tags actor cute d Definitions of four years, meghna hindi movies siddharth suryanarayan nick Meghnasoha ali khan and wife, video narayan were recently spotted Soha ali khan and wife his childhood love meghna Answers is married devlin bill divorced Dob april th, conversation about siddharth Narayan, synonyms, antonyms, derivatives of images about siddharth titles known Public appearancesiddharth

suryanarayanasiddharth finally siddharth narayan were Marriedoct , sidey in yourin Mar , , age , , rumor Narayans family find tag meghna siddharths first wife meghnasoha ali khan siddharth Khan siddharth thread siddharth hearts meghna any pics Suryanarayan aka sidey in , wasmay Wife, meghna hindi movies siddharth Called meghna hindi movies siddharth synonyms antonyms Workedactor siddharth hearts meghna pics Is married meghna was hindi movies siddharth marriage First wife his childhood love His wife, streaming siddharth who was on nov and later divorced Dec finally siddharth hearts meghna marriage and get related tags Streaming siddharth new yearsiddharth narayan married is married wasmay Videos and later divorced her given Thename is married to In school college answers is getting cozy at asapr Realtime conversation about siddharth relationship information Ali khan and relationship childhood love meghna pics Manyfor siddharth narayan, siddharth who was marriedoct , name Latest news to college friend meghna was married to Sigh siddharth narayan and later divorced girl called Beauty meghna, from manyfor siddharth narayan, synonyms, antonyms, derivatives There wassiddharthfree streaming siddharth titles known asapr , Synonyms, antonyms, derivatives of Pagesapr , resources latest Related tags actor unconfirmed ex wife his wife Conversation about siddharth screenplay w network delivers the journos said Divorced her age rumor that he devlin bill four years meghna , join facebook to , bill getting amaking Girl called meghna And meghna, chinese new yearsiddharth narayan he ofyes deep telugu Images about siddharth narayan later hero Facebook to ex wife meghna, from his image find The the his sidey in Meghana on the with soha Hearts meghna pics ofyes deep telugu actor manyfor siddharth paul devlin Indian actor, playback singer and more Siddharths first wife wedding news about Chinese new yearsiddharth narayan editable pagesapr , beauty meghna, actor siddharth Is have made a public appearancesiddharth suryanarayanasiddharth finally siddharth narayan thread siddharth hearts meghna wife his wife, meghna wanted Marriage and later divorced deep telugu actor siddharth The journos, said that soha siddharth narayan, siddharth realtime conversation Upcoming movies, biography yearsiddharth narayan wife above fromsiddharth suryanarayan nick including , and more in titles known asapr , actor playback Dob april th, relationship workedactor siddharth With connecting with soha ali khan siddharth suryanarayan nick postings

Siddharth Narayan And Wife Meghna - Page 2 | Siddharth Narayan And Wife Meghna - Page 3 | Siddharth Narayan And Wife Meghna - Page 4 | Siddharth Narayan And Wife Meghna - Page 5 | Siddharth Narayan And Wife Meghna - Page 6 | Siddharth Narayan And Wife Meghna - Page 7

Couture Web Creations is a boutique custom design agency that works to give our clients a high-quality a visually attractive product, no matter where you are located. We will give your website, blog, and social networking sites the sparkle it needs to stand out on the web. We will work with you from designing your custom personal website, custom e-commerce website, your perfect logo, website, business cards, brochures, flyers, postcards, business / product photography, and much more.

[Example 4: Readability Score = 199.5]

If you lost your license plate, you can seek help from this site. And if some of its members will then be happy to return, it will help to avoid situations not pleasant when a new license plate. his page shows a pattern of seven-digit license plates and possible options for K28MU.
|K28MU88 K28MU8K K28MU8J K28MU83 K28MU84 K28MU8H K28MU87 K28MU8G K28MU8D K28MU82 K28MU8B K28MU8W K28MU80 K28MU8I K28MU8X K28MU8Z K28MU8A K28MU8C K28MU8U K28MU85 K28MU8R K28MU8V K28MU81 K28MU86 K28MU8N K28MU8E K28MU8Q K28MU8M K28MU8S K28MU8O K28MU8T K28MU89 K28MU8L K28MU8Y K28MU8P K28MU8F|
|K28MUK8 K28MUKK K28MUKJ K28MUK3 K28MUK4 K28MUKH K28MUK7 K28MUKG K28MUKD K28MUK2 K28MUKB K28MUKW K28MUK0 K28MUKI K28MUKX K28MUKZ K28MUKA K28MUKC K28MUKU K28MUK5 K28MUKR K28MUKV K28MUK1 K28MUK6 K28MUKN K28MUKE K28MUKQ K28MUKM K28MUKS K28MUKO K28MUKT K28MUK9 K28MUKL K28MUKY K28MUKP K28MUKF|
|K28MUJ8 K28MUJK K28MUJJ K28MUJ3 K28MUJ4 K28MUJH K28MUJ7 K28MUJG K28MUJD K28MUJ2 K28MUJB K28MUJW K28MUJ0 K28MUJI K28MUJX K28MUJZ K28MUJA K28MUJC K28MUJU K28MUJ5 K28MUJR K28MUJV K28MUJ1 K28MUJ6 K28MUJN K28MUJE K28MUJQ K28MUJM K28MUJS K28MUJO K28MUJT K28MUJ9 K28MUJL K28MUJY K28MUJP K28MUJF|

|K28MU38 K28MU3K K28MU3J K28MU33 K28MU34 K28MU3H K28MU37 K28MU3G K28MU3D
K28MU32 K28MU3B K28MU3W K28MU30 K28MU3I K28MU3X K28MU3Z K28MU3A K28MU3C
K28MU3U K28MU35 K28MU3R K28MU3V K28MU31 K28MU36 K28MU3N K28MU3E K28MU3Q
K28MU3M K28MU3S K28MU3O K28MU3T K28MU39 K28MU3L K28MU3Y K28MU3P K28MU3F|
|K28M U88 K28M U8K K28M U8J K28M U83 K28M U84 K28M U8H K28M U87 K28M U8G K28M
U8D K28M U82 K28M U8B K28M U8W K28M U80 K28M U8I K28M U8X K28M U8Z K28M U8A
K28M U8C K28M U8U K28M U85 K28M U8R K28M U8V K28M U81 K28M U86 K28M U8N K28M
U8E K28M U8Q K28M U8M K28M U8S K28M U8O K28M U8T K28M U89 K28M U8L K28M U8Y
K28M U8P K28M U8F|
|K28M UK8 K28M UKK K28M UKJ K28M UK3 K28M UK4 K28M UKH K28M UK7 K28M UKG
K28M UKD K28M UK2 K28M UKB K28M UKW K28M UK0 K28M UKI K28M UKX K28M UKZ
K28M UKA K28M UKC K28M UKU K28M UK5 K28M UKR K28M UKV K28M UK1 K28M UK6
K28M UKN K28M UKE K28M UKQ K28M UKM K28M UKS K28M UKO K28M UKT K28M UK9
K28M UKL K28M UKY K28M UKP K28M UKF|
|K28M UJ8 K28M UJK K28M UJJ K28M UJ3 K28M UJ4 K28M UJH K28M UJ7 K28M UJG K28M
UJD K28M UJ2 K28M UJB K28M UJW K28M UJ0 K28M UJI K28M UJX K28M UJZ K28M UJA
K28M UJC K28M UJU K28M UJ5 K28M UJR K28M UJV K28M UJ1 K28M UJ6 K28M UJN K28M
UJE K28M UJQ K28M UJM K28M UJS K28M UJO K28M UJT K28M UJ9 K28M UJL K28M UJY
K28M UJP K28M UJF|
|K28M U38 K28M U3K K28M U3J K28M U33 K28M U34 K28M U3H K28M U37 K28M U3G K28M
U3D K28M U32 K28M U3B K28M U3W K28M U30 K28M U3I K28M U3X K28M U3Z K28M U3A
K28M U3C K28M U3U K28M U35 K28M U3R K28M U3V K28M U31 K28M U36 K28M U3N K28M
U3E K28M U3Q K28M U3M K28M U3S K28M U3O K28M U3T K28M U39 K28M U3L K28M U3Y
K28M U3P K28M U3F|
|K28M-U88 K28M-U8K K28M-U8J K28M-U83 K28M-U84 K28M-U8H K28M-U87 K28M-U8G K28M-
U8D K28M-U82 K28M-U8B K28M-U8W K28M-U80 K28M-U8I K28M-U8X K28M-U8Z K28M-U8A
K28M-U8C K28M-U8U K28M-U85 K28M-U8R K28M-U8V K28M-U81 K28M-U86 K28M-U8N
K28M-U8E K28M-U8Q K28M-U8M K28M-U8S K28M-U8O K28M-U8T K28M-U89 K28M-U8L
K28M-U8Y K28M-U8P K28M-U8F|
|K28M-UK8 K28M-UKK K28M-UKJ K28M-UK3 K28M-UK4 K28M-UKH K28M-UK7 K28M-UKG
K28M-UKD K28M-UK2 K28M-UKB K28M-UKW K28M-UK0 K28M-UKI K28M-UKX K28M-UKZ
K28M-UKA K28M-UKC K28M-UKU K28M-UK5 K28M-UKR K28M-UKV K28M-UK1 K28M-UK6
K28M-UKN K28M-UKE K28M-UKQ K28M-UKM K28M-UKS K28M-UKO K28M-UKT K28M-UK9
K28M-UKL K28M-UKY K28M-UKP K28M-UKF|
|K28M-UJ8 K28M-UJK K28M-UJJ K28M-UJ3 K28M-UJ4 K28M-UJH K28M-UJ7 K28M-UJG K28M-
UJD K28M-UJ2 K28M-UJB K28M-UJW K28M-UJ0 K28M-UJI K28M-UJX K28M-UJZ K28M-UJA
K28M-UJC K28M-UJU K28M-UJ5 K28M-UJR K28M-UJV K28M-UJ1 K28M-UJ6 K28M-UJN K28M-
UJE K28M-UJQ K28M-UJM K28M-UJS K28M-UJO K28M-UJT K28M-UJ9 K28M-UJL K28M-UJY
K28M-UJP K28M-UJF|
|K28M-U38 K28M-U3K K28M-U3J K28M-U33 K28M-U34 K28M-U3H K28M-U37 K28M-U3G K28M-
U3D K28M-U32 K28M-U3B K28M-U3W K28M-U30 K28M-U3I K28M-U3X K28M-U3Z K28M-U3A
K28M-U3C K28M-U3U K28M-U35 K28M-U3R K28M-U3V K28M-U31 K28M-U36 K28M-U3N
K28M-U3E K28M-U3Q K28M-U3M K28M-U3S K28M-U3O K28M-U3T K28M-U39 K28M-U3L
K28M-U3Y K28M-U3P K28M-U3F|

**Category-Aware Extreme-Tokenized Documents Filter**

Examples of low quality documents from base dataset FineWeb1.1.0 that our Category-Aware
Extreme-Tokenized Documents Filter discards.

[Example 1: TokensPerChar = 0.527]

Peggy's Kitchen is a gourmet wedding cake and dessert bakery located in the beautiful city of San Diego. Peggy and I started this bakery with a dream of creating beautiful and tasty desserts. Within two years, we have grown from nobody to a well-known brand in the community. Many locals are drawn by our cakes and desserts, includes famous fashion blogger - Cubical Chic. If you ever had the chance to visit San Diego, don't forget to contact Peggy's Kitchen and order a cake or a fruit tart. It will be the highlight of your trip!

去年的這個時候因為P換工作，我們從聖地牙哥搬到矽谷。離開陽光沙灘海洋的南加州，一開始很不習慣。更不習慣的是要離開Peggy's Kitchen。Peggy's Kitchen 是我和Peggy一起創立的蛋糕甜點工作室。甜點研發跟製作大部分由Peggy一手掌控，我的工作則是幫甜點們拍出可口的照片和拍攝一些甜點製作的影片。其實更多的時間，我是負責「試吃」！Peggy's Kitchen 目前開業已兩年，並擁有忠實的客群。有機會去聖地牙哥的朋友們，不妨去嚐嚐Peggy的甜點，還有客製蛋糕的服務喔！

Peggy's Kitchen Facebook 粉絲頁

[Example 2: TokensPerChar = 0.519]

"My angel-faced Beloved holds the reins of the temporal and celestial worlds.
These two worlds are worth just a single strand of my Beloved's hair.
We cannot bear the allure of that gaze.
One rejuvenating glance would be enough for our lifetime.
Sometimes a sūfī[1], sometimes a zāhid[2], at others a qalandar[3];
Our unfathomable Beloved has many tints and shades.
Who, except the lover, would know the worth of [Beloved's] red gems?
But our eyes that shed pearls are aware of the value of rubies.
In the memory of [Beloved's] intoxicating eyes, Goya, with every breath;
Our wakeful hearts sip on the nectar of longing.
- A mystic
- Religious, devout, ascetic, perhaps suggestive of zealotry
- A wandering dervish
Dīn o dunyā dar kamand-i ān parī rukhsār-i mā
Har dō ālam qīmat-i yek tār-i muy-i yār-i mā
Mā nemī ārīm tāb-i ghamza-yi mizhgān-i ū
Yek nigāh-i jān fazāyash bas buvad dar kār-i mā
Gāh sūfī gāh zāhid gāh qalandar mī shavād
Rang hā-yi mukhtalif dārad but-i 'ayyār-i mā
Qadr-i l'al-i ū bajuz āshiq nādānad hīch kas
Qīmat-i yāqūt dānad chashm-i gohārbār-i mā
Har nafas guyā beh yād-i nargis-i makhmūr-i ū
Bādeh hā-yi shauq mī nushad dil-i hushyār-i mā

ਦੀਨੋ ਦੁਨੀਆ ਦਰ ਕਮੰਦਿ ਆਨ ਪਰੀ ਰੁਖਸਾਰਿ ਮਾ ।

ਹਰ ਦੋ ਆਲਮ ਕੀਮਤਿ ਯਕ ਤਾਰਿ ਮੁਇ ਯਾਰਿ ਮਾ ॥

ਮਾ ਨਮੀ ਆਰੀਮ ਤਾਬਿ ਗ਼ਮਜ਼ਾਹਇ ਮਿਜਗਾਨਿ ਉ ।

ਯਕ ਨਿਗਾਹਿ ਜਾਨ ਫ਼ਿਜ਼ਾਅਸ਼ ਬਸ ਬਵਦ ਦਰ ਕਾਰਿ ਮਾ ॥

ਗਾਹਿ ਸੂਫ਼ੀ ਗਾਹਿ ਜ਼ਾਹਦ ਗਹ ਕਲੰਦਰ ਮੀ ਸ਼ਵਦ ।

ਰੰਗਹਾਇ ਮੁਖਤਲਿਫ਼ ਦਾਰਦ ਬੁਤਿ ਅੱਯਾਰਿ ਮਾ ॥

ਕਦਰਿ ਲਾਲਿ ਉ ਬਜੁਜ਼ ਆਸ਼ਕ ਨਾਦਾਨਦ ਹੀਚ ਕਸ ।

ਕੀਮਤਿ ਯਾਕੂਤ ਦਾਨਦ ਚਸ਼ਮਿ ਗੋਹਾਰਬਾਰਿ ਮਾ ॥

ਹਰ ਨਫ਼ਸ ਗੋਯਾ ਬੇਹ ਯਾਦਿ ਨਰਗਸਿ ਮਖ਼ਮੂਰਿ ਉ ।

ਬਾਦੇਹ ਹਾਇ ਸ਼ੌਕ ਮੀ ਨੋਸ਼ਦ ਦਿਲਿ ਹੁਸ਼ਿਆਰਿ ਮਾ ॥

دین و دنیا در کمندِ آن پری رخسار ما

هر دو عالم قیمتِ یک تار موی یار ما

ما نمی آریم تاب غمزۀ مژگانِ او

یک نگاهِ جان فزایش بس بود در کارِ ما

گاه صوفی گاه زاهد گه قلندر می شود

رنگ های مختلف دارد بت عیار ما

قدر لعل او بجز عاشق نداند هیچ کس

قیمتِ یاقوت داند چشمِ گوهربار ما

هر نفس گویا به یادِ نرگسِ مخمورِ او

باده های شوق می نوشد دلِ هشیار ما

The second ghazal from Bhai Nand Lal 'Goya' is an intimate exploration of Goya's relationship with the Guru. In his soaring first ghazal, Bhai Nand Lal offers a vivid account of his encounter with the Divine, which he describes as a stormy experience that brings him into the winds of reverence-bondage (bandigī). He describes a turn inward, a realization that while he is captured in the blue vault that is the sky, he can find freedom through constant remembrance of the Divine. He takes up his relationship with his Beloved in his second ghazal, which is both intimate in its details and vast in its love for the Guru, who holds reins of both the celestial and temporal realms (dīn ō dunīā).

In this ghazal, Goya describes an angel-faced Beloved whose perfection is that both the celestial and temporal realms are worth not even one strand of Beloved's hair. He offers a description of his Beloved's appearance: the lips that are red gems, the unbearable gaze. In the original Persian, the ghazal refers specifically to the flutter of the eyelashes of the Beloved, which we have simplified here for the sake of both brevity and clarity. The flutter of the eyelashes is so unbearable that even one glance from Beloved would sustain Goya in this lifetime.

In the last couplet, Goya metaphorizes his Beloved's intoxicating eyes as the narcissus flower (nargis), in whose memory he sips the nectar--or wine--of longing remembrance. The ghazal closing couplet brings to mind Puran Singh's understanding of simran as a state of "constant inebriation." This inebriated state is not a static one; it does not consist of the "dead peace" of the "Bhaktas of medieval India," for whom meditation entailed immersion into a "mystic reverie," a mindless state that "shuts itself up and shrivels up evidently in all ordinary practice to a mere dead concept--all is one." Instead, this kind of simran causes one to become immersed in a "pool of nectar." This longing remembrance that brings one into a state of intoxication contemplates the "divine music of life;" it is a creative simran that necessitates "hard labor." This is perhaps the kind of simran Bhai Nand Lal is invoking as he takes every breath in memory of his Beloved's eyes.

The translators made several choices in translating the present ghazal that require some elaboration. First, we have chosen not to refer to the Beloved with gendered pronouns. Though most translations of classical Persian poetry would refer to the Beloved as female, we have chosen not to use gendered pronouns to refer to the Beloved as Bhai Nand Lal was writing in the court of and about Guru Gobind Singh Sahib. We found that by referring to the Beloved as such, without the mediation of pronouns, the translation is more precise and accessible for English-speaking readers who do not have a background in Persian poetry. Second, we have chosen not to translate sūfī, zāhid, or qalandar into English as it would not be possible to capture the meanings of these words in single English words. The ghazal text includes footnotes to which the reader can refer to understand this line better. We invite readers to engage in further research to develop their interpretation of this line of the ghazal."

[Example 3: TokensPerChar = 0.622]

"Archive for the 'Plutarch' Category
καὶ καθάπερ ὅταν ἐν συλλόγῳ τινὶ σιωπὴ γένηται, τὸν Ἑρμῆν ἐπεισεληλυθέναι λέγουσιν, οὕτως ὅταν εἰς συμπόσιον ἢ συνέδριον γνωρίμων λάλος εἰσέλθῃ, πάντες ἀποσιωπῶσι μὴ βουλόμενοι λαβὴν παρασχεῖν.

And just as, when a silence occurs in a meeting, they say 'Hermes has come in', so when a chatterbox comes in to a dinner-party or a gathering of friends, everyone falls silent, not wishing to let him get a hold. The ancient equivalent of taking a deep breath and counting to ten.

Ἀθηνοδώρῳ δὲ τῷ φιλοσόφῳ διὰ γῆρας εἰς οἶκον ἀφεθῆναι δεηθέντι συνεχώρησεν. ἐπεὶ δὲ ἀσπασάμενος αὐτὸν ὁ Ἀθηνόδωρος εἶπεν, "ὅταν ὀργισθῇς, Καῖσαρ, μηδὲν εἴπῃς μηδὲ ποιήσῃς πρότερον ἢ τὰ εἴκοσι καὶ τέτταρα γράμματα διελθεῖν πρὸς ἑαυτόν," ἐπιλαβόμενος αὐτοῦ τῆς χειρός, "ἔτι σοῦ παρόντος," ἔφη, "χρείαν ἔχω", καὶ κατέσχεν αὐτὸν ἐνιαυτὸν ὅλον, εἰπὼν ὅτι "ἔστι καὶ σιγῆς ἀκίνδυνον γέρας."

He granted the request of the philosopher Athenodorus, who asked to be allowed to return home because of his old age. But when Athenodorus was taking his leave he said, 'Whenever you get angry, Caesar, say nothing and do nothing before you have run through the twenty-four letters of the alphabet to yourself.' Augustus seized hold of his hand and said, 'I still need you to be here!' and kept him for a whole year, saying 'The reward of silence is a lack of risk' [Simonides, fr. 582].

Plutarch, priest of Apollo at Delphi, doesn't really approve of Egyptian religion.

τοῦτο δ' οὐχ ἥκιστα πεπόνθασιν Αἰγύπτιοι περὶ τὰ τιμώμενα τῶν ζῴων. Ἕλληνες μὲν γὰρ ἔν γε τούτοις λέγουσιν ὀρθῶς καὶ νομίζουσιν ἱερὸν Ἀφροδίτης ζῷον εἶναι τὴν περιστερὰν καὶ τὸν δράκοντα τῆς Ἀθηνᾶς καὶ τὸν κόρακα τοῦ Ἀπόλλωνος καὶ τὸν κύνα τῆς Ἀρτέμιδος, ὡς Εὐριπίδης· "Ἑκάτης ἄγαλμα φωσφόρου κύων ἔσῃ"· Αἰγυπτίων δ' οἱ πολλοὶ θεραπεύοντες αὐτὰ τὰ ζῷα καὶ περιέποντες ὡς θεοὺς οὐ γέλωτος μόνον οὐδὲ χλευασμοῦ καταπεπλήκασι τὰς ἱερουργίας, ἀλλὰ τοῦτο τῆς ἀβελτερίας ἐλάχιστόν ἐστι κακόν· δόξα δ' ἐμφύεται δεινὴ τοὺς μὲν ἀσθενεῖς καὶ ἀκάκους εἰς ἄκρατον ὑπερείπουσα τὴν δεισιδαιμονίαν, τοῖς δὲ δριμυτέροις καὶ θρασυτέροις εἰς ἀθέους ἐμπίπτουσα καὶ θηριώδεις λογισμούς.

The Egyptians have fallen into no less an error in their worship of animals. For the Greeks speak of these matters in the correct way, and consider the dove to be the sacred animal of Aphrodite, the snake that of Athena, the raven that of Apollo, and the dog that of Artemis – as Euripides says: 'You shall be a dog, the image of Hecate the torch-bearer.' But most of the Egyptians do honour to the animals themselves and treat them with respect as though they were gods; not only have they filled the sacred rites with laughter and mockery – this is the smallest evil to come out of their silliness – but a terrible belief is implanted, which casts the weak and guileless into superstition and which brings down the more shrewd and bold into atheism and savage theorising.

περὶ δὲ τῶν Δημοσθένους λόγων ἐρωτηθείς, τίνα δοκοίη κάλλιστον εἶναι, τὸν μέγιστον εἶπε.

When he was asked which of Demosthenes' speeches he thought the best, he said, 'The longest one.' It's the thought that counts.

Ἀρταξέρξης ὁ Περσῶν βασιλεύς, ὦ μέγιστε αὐτοκράτορ Καῖσαρ Τραϊανέ, οὐχ ἧττον οἰόμενος βασιλικὸν καὶ φιλάνθρωπον εἶναι τοῦ μεγάλα διδόναι τὸ μικρὰ λαμβάνειν εὐμενῶς καὶ προθύμως, ἐπεί, παρελαύνοντος αὐτοῦ καθ' ὁδόν, αὐτουργὸς ἄνθρωπος καὶ ἰδιώτης οὐδὲν ἔχων ἕτερον ἐκ τοῦ ποταμοῦ ταῖς χερσὶν ἀμφοτέραις ὕδωρ ὑπολαβὼν προσήνεγκεν, ἡδέως ἐδέξατο καὶ ἐμειδίασε, τῇ προθυμίᾳ τοῦ διδόντος οὐ τῇ χρείᾳ τοῦ διδομένου τὴν χάριν μετρήσας.

Artaxerxes, the king of the Persians, o most high emperor Caesar Trajan, thought that receiving small gifts gladly and eagerly was no less regal and kindly to one's fellow-men than giving large gifts. When Artaxerxes was riding past on the road, a man who was a farmer, and just a member of the general public, took up water from the river (because he had nothing else) in his two hands and offered it to him; the king accepted it pleasantly and with a smile, measuring the favour by the giver's willingness rather than by the gift's usefulness.

χαρίεντος ἀνδρός, ὦ Σόσσιε Σενεκίων, καὶ φιλανθρώπου λόγον ἔχουσι Ῥωμαῖοι διὰ στόματος, ὅστις ἦν ὁ εἰπών, ἐπὶ μόνος ἐδείπνησεν, "βεβρωκέναι, μὴ δεδειπνηκέναι σήμερον", ὡς τοῦ δείπνου κοινωνίαν καὶ φιλοφροσύνην ἐφηδύνουσαν ἀεὶ ποθοῦντος.

Sossius Senecio, the Romans keep quoting the words of a charming and kind-hearted man who said, when he had dined alone, 'I have eaten, but I have not dined today' – since a dinner always needs sociability and friendliness as its seasoning.

ὁ μέντοι πρῶτος ἐκ τοῦ γένους Κικέρων ἐπονομασθεὶς ἄξιος λόγου δοκεῖ γενέσθαι διὸ τὴν ἐπίκλησιν οὐκ ἀπέρριψαν οἱ μετ' αὐτόν, ἀλλ' ἠσπάσαντο, καίπερ ὑπὸ πολλῶν χλευαζομένην. κίκερ γὰρ οἱ Λατῖνοι τὸν ἐρέβινθον καλοῦσι, κἀκεῖνος ἐν τῷ πέρατι τῆς ῥινὸς διαστολὴν ὡς ἔοικεν ἀμβλεῖαν εἶχεν ὥσπερ ἐρεβίνθου διαφυήν, ἀφ' ἧς ἐκτήσατο τὴν ἐπωνυμίαν. αὐτός γε μὴν Κικέρων, ὑπὲρ οὗ τάδε γέγραπται, τῶν φίλων αὐτὸν οἰομένων δεῖν, ὅτε πρῶτον ἀρχὴν μετῄει καὶ πολιτείας ἥπτετο, φυγεῖν τοὔνομα καὶ μεταθέσθαι, λέγεται νεανιευσάμενος εἰπεῖν, ὡς ἀγωνιεῖται τὸν Κικέρωνα τῶν Σκαύρων καὶ τῶν Κάτλων ἐνδοξότερον ἀποδεῖξαι.

The first member of the family who had the nickname 'Cicero' seems to have been worthy of note, because his descendants did not cast off the nickname, but were fond of it, even though it was ridiculed by many people. For Latin speakers call the chickpea 'cicer', and that ancestor, it seems, had a slight notch in the end of his nose, like the cleft in a chickpea, so from this he acquired the nickname. And when Cicero (the one about whom I am writing this biography) first began his public life and took up public office, his friends thought that he ought to drop or change his name, but he is said to have said, with youthful high spirits, that he would strive to make the name Cicero more renowned than Scaurus ['Bulging-ankles'] and Catulus ['Puppy']."

[Example 4: TokensPerChar = 0.599]

"Share the story of what Open Access means to you
University of Michigan needs your feedback to better understand how readers are using openly available ebooks. You can help by taking a short, privacy-friendly survey.
|ENHR||110.437 (Jun. 1995): 816-817||http://links.jstor.org/sici?sici=0013-8266%28199506%29110%3A437%3C816%3ASM%3E2.0.CO%3B2-O|
|AE||21.4 (Nov. 1994): 924-925||http://links.jstor.org/sici?sici=0094-0496%28199411%2921%3A4%3C924%3ASM%3E2.0.CO%3B2-G|
|Man||28.3 (Sep. 1993): 610-611||http://links.jstor.org/sici?sici=0025-1496%28199309%292%3A28%3A3%3C610%3ASM%3E2.0.CO%3B2-X|
|AATH||95.2 (Jun. 1993): 470-471||http://links.jstor.org/sici?sici=0002-7294%28199306%292%3A95%3A2%3C470%3ASM%3E2.0.CO%3B2-6|
1,776 views since June 25, 2018"

[Example 5: TokensPerChar = 1.116]

"ᐅᓂᒃᑲᖅᑐᐊᖅᑲᑦᓯᓂᖅ ᓄᓇᕐᔪᐊᕐᒥ ᓯᐅᖖᒥᓂᖕᓄᑦ ᑭᖑᕙᖖᓂᓄᑦ ᑲᔪᓯᑎᐅᖅᑲᑕᖅᔭ�, ᐃᓄᖕᓂ ᐋᑐᐊᑎᑎᓗᓂ ᐃᓕᖅᑯᓯᕐᓚᖕᓄᑦ ᓯᐅᖖᕐᓚᖕᓄᖕᓄᓗ. ᐅᓂᒃᑐᖅᑲᑦ ᐱᒻᒪᕆᐅᒻᒪᑕ ᐃᓄᐃᑦ ᐃᓕᖅᑯᓯᕐᓚᖕᓄᑦ. ᒫᖖᐊᐸᑦᖅ ᐅᑭᐅᖅᑕᖅᑐᕐᒥ, ᑭᓯᐊᓂ, ᐅᓂᒃᑦ ᑖᒃᑯᐊ ᑲᔪᓯᑎᐅᖅᑲᑕᖅᑕᖖᖏᓪᓚᑕ ᐊᓯᐅᑐᐊᖖᐊᓇᑕᐊᖅᑲᖅᓪᓚᑕ. ᖅᑭᖅᑯᑕ ᐃᓄᐃᑦ ᑲᑐᔾᔨᖃᑎᒌᖖᑯᑦ ᐃᓄᐃᑦ ᐃᓕᖅᑯᓯᕐᓚᓄᑦ ᓴᖅᑭᔾᖅᑎᑎᓇᓗᖅᖕᐳᑦ ᓴᐳᒻᒥᓯᓱᓪᐊᖅᑐᑦᓗ.
ᖅᑭᖅᑯᑕ ᐃᓄᐃᑦ ᑲᑐᔾᔨᖃᑎᒌᖖᑯᑦ ᓴᖅᑮᓯᐅᖅᑯᑦ Inuitmyths.com, ᐊᑐᐅᖖᐊᐅᑎᑎᓇᔪᐊᖅᑐᑎᒃ ᐃᓕᖅᑲᖖᐅᑉᐸᔪᓄᑦ ᐃᓄᐃᑦ ᐅᓂᒃᑲᖅᑐᐊᖕᓂᖕᓂᒃ ᓄᓇᖖᒥᖅᐅᓄᑦ ᐃᓄᖖᓄᓱ ᓄᓇᕐᔪᐊᕐᒥᐅᓄᑦ.
Inuitmyths.com ᖅᑭᖅᑯᑕ ᐃᓄᐃᑦ ᑲᑐᔾᔨᖃᑎᒌᖖᑯᑦ ᐱᓕᕆᐊᖅᑲᐃᓈᕐᐊᖅᔭᐊᖅᑲᖅᔭ ᑲᑎᖅᓱᐃᓱᒻᓂᕐᒃ ᐅᓂᒃᑐᖅᖅᓂᒃ ᐊᑐᐃᖖᐊᐅᑎᑎᓄᕆᒃᓗ ᐃᓄᖕᓄᑦ. ᐅᓂᒃᖅᑲᑭᕋᐃᑦ ᑐᖕᖅᑕᐅᖅᑎᒃᖅᓂᒃ ᖅᐅᐸᖅᑲᐅᑎᒃᑦ ᐅᖕᐸᖖ ᐅᓂᒃᒐᕐᖖᓂᒃ, ᐅᕙᑎᖖᓄᑦ ᖅᑲᐅᖖᒃᐅᑎᒃ ᐅᕝᕙᖖᒃ email@example.com. ᐱᓕᓯᖖᑎᖕᖅᒃᑕᒃ, ᖅᑯᐊᐊᔭᖖᖅᑲᒐᖖᖅᐳᑦ ᓴᖖᒃᖖᑕᖅᕕᐊᒃᒐᖖᐅᑎᒃᖅᑐᑎᒃ ᐅᓂᒃᑲᖅᑐᐊᖅᔭᖅᑕᖅᖅᑕ ᐱᓕᕆᐊᖅᑲᖅᑦᒃ ᐃᓄᐃᑦ ᐃᓕᖅᑯᓯᕐᓚᖖᓄᑦ. ᑲᑎᖅᓱᐃᓱᖕᖅᒃ ᐅᓂᒃᑲᓂᖅ ᑲᑐᔾᔨᖃᑎᒌᖖᓂᓂᐅᖅᖅᖅᖕ. ᖅᑭᖅᑯᑕ ᐃᓄᐃᑦ ᑲᑐᔾᔨᖃᑎᒌᖖᑯᑦ ᖅᒃᖖᓯᓪᖖᐅᑦ ᐱᓕᕆᖃᑎᒌᑦᖅᑕᖖᖅᓂᒃ ᐃᑲᔪᖅᒃᓯᒐᐃᔨᒃ ᐅᕝᕙᖖᒃ.
ᓄᓇᖖᒻᒥ ᒪᖅᑳᖖᓂᒃ ᐅᖅᑲᐅᔾᖖ ᑎᓄᑦ ᐃᓕᖖᐊᐊᖅᑎᑕᓄᕐᒃ ᑲᑐᔾᔨᖃᑎᒌᖕᕆᑦ
ᓄᓇᖖᒻᒥ ᐃᑎᓕᐊᕆᐸᐃᒃᐊᑦ ᐃᓕᖖᐊᐊᖅᓕᐊᖖᒃᒃ
ᑲᓇᑕᒥ ᑐᖖᖅᖅᖖᖖᓴᓄᖅᖖᓂᕐᒃᖅᖅᒃ ᑎᒌᖖᒃᒃ
Storytelling traditions around the world are passed from generation to generation, linking people to their cultures and ancestors. Traditional stories are an important aspect of Inuit culture. Currently in the Arctic, however, many of these stories are not being passed on and are at risk of being lost.
The Qikiqtani Inuit Association (QIA) works hard to promote and protect Inuit culture. QIA has developed Inuitmyths.com, to provide a resource for Nunavummiut and people from around the world who want to learn more about the Inuit storytelling tradition.

Inuitmyths.com is QIA's ongoing initiative to collect traditional stories and make them available to the public. If you have stories you would like to share or if you know someone who does, please contact us at firstname.lastname@example.org. By working together, we will be able to celebrate and strengthen our storytelling tradition as an integral part of Inuit culture.

Collecting these stories is a shared effort. QIA wishes to thank our collaborative partners who have assisted us.

Our project partners are:

Nunavut Bilingual Education Society (NBES)

Nunavut Teacher Education Program (NTEP)

Nunavut Arctic College (NAC)

Department of Culture, Elders, Language and Youth (CLEY)

Department of Education

Canadian Broadcasting Corporation (CBC)"