# OpenReview forum: "GneissWeb: Preparing High Quality Data for LLMs at Scale"
_ICLR.cc/2026/Conference — ICLR 2026 Poster_

### Official Review · Reviewer_EiPH · 2025-10-31

**Soundness:** 3
**Presentation:** 3
**Contribution:** 3
**Rating:** 6
**Confidence:** 3

**Summary:**

This method uses an ensemble of quality filters and exact deduplication to improve the pretraining data recipe. This paper works, demonstrates consistent improvement across scales. The evaluations were diverse and rigorous.

**Strengths:**

1. Powerful empirical results across the benchmarks. I really like the scaling plots. Tell a very clear story.
2. The mixing of multiple quality filters is a nice paradigm.
3. The evaluations were particularly rigorous.
4. Good statistical rigor.

**Weaknesses:**

None really. Small things:
1. Could have discussed the computational cost of experiments.
2. Starting from FineWeb is good, but maybe could have ablated different sources.
3. Math/code performance is ok, but not stellar.

**Questions:**

N/A

---

> ### Author Response · Authors · 2025-11-22
> **Addressing Compute Cost, Data Source Choices, and Domain Performance (Rebuttal 1/1)**
>
> We thank the reviewer for highlighting the strength of our empirical results across diverse benchmarks, the clarity of our scaling analyses, the novelty of combining multiple quality filters, and the rigor of our evaluations. We address their comments below.
>
> 1. **Cost Details**
>
> - Estimating the *total* computational cost of all experiments is inherently challenging due to several failures during development as well as failed runs due to cluster issues, data processing errors (e.g., human mistakes, logging/database failures), and other transient problems. However, we will include a rough estimate of compute cost for ablation experiments in the revised manuscript.
>
> - For dataset creation, we recorded resource consumption, which we summarize below (processing ~4B documents, ~4.5 TB). We will incorporate these details in the final version.
>
> | Recipe Ingredient              | # of replicas | # of CPUs per replica | Memory per replica (GB) | Time (hh:mm:ss) |
> |--------------------------------|---------------|------------------------|--------------------------|-----------------|
> | Exact Substring Deduplication  | 30            | 8                      | 80                       | 25:57:03        |
> | fastText quality annotations   | 120           | 8                      | 100                      | 01:19:49        |
> | fasttext Category Annotation   | 10            | 16                     | 160                      | 28:02:25        |
> | Readability Score Annotation   | 30            | 16                     | 160                      | 04:00:28        |
> | Tokenization                   | 70            | 4                      | 64                       | 02:15:44        |
> | Extreme-Tokenized Annotation   | 20            | 16                     | 160                      | 00:40:49        |
>
> 2. **Starting with Different Data Sources**
>
> - Our primary goal is to design **a data curation recipe suitable for Stage-1 training**, which requires massive scale and broad coverage.
> - Leading LLMs typically train on tens of trillions of tokens (e.g., Llama-3 on ~15T tokens, Gemma-2 on ~13T tokens,  SmolLM3 on ~11T tokens). Pre-training routinely involves multiple stages: in the first stage, the model is trained on a very large, diverse corpus to maximize coverage, followed by subsequent stages with progressively higher quality data.
> - Common Crawl remains the dominant source for Stage-1 data. Even aggressively filtered datasets such as DCLM-Baseline and FineWeb-Edu originate from Common Crawl. We started GneissWeb on top of FineWeb, which itself derives from the Common Crawl.
> - An ideal comparison between the datasets would involve pre-training a model on ~10T to 12T tokens using GneissWeb in Stage-1 versus alternatives (e.g., FineWeb or RedPajamaV2, or upsampling high-quality smaller datasets such as DCLM-Baseline). Unfortunately, experiments of this scale are not feasible within our compute budget. We will acknowledge this limitation explicitly in the revised manuscript.
> - We note that many influential dataset papers (e.g., Dolma, RefinedWeb, FineWeb) conduct experiments on 100B-350B tokens, which is the scale we follow. Therefore, our experimental scale (100B–350B tokens) aligns with several dataset papers (Dolma, RefinedWeb, FineWeb).
>
> 3. **Math/Code Performance**
>
> - Our focus is on Stage-1 pre-training, prioritizing large-scale coverage rather than specialized domains.
>
> - It has been observed in the literature that performance on Code and Math during Stage-1 is typically modest, as these capabilities are significantly enhanced in later stages with higher quality data.
>
> - For example, DCLM-Baseline explicitly incorporates ProofPile2 (Math data) and StarCoder (Code data) in subsequent phases to improve math and code. Our results are consistent with this trend and will be clarified in the revised manuscript.

---

### Official Review · Reviewer_c8pE · 2025-11-01

**Soundness:** 2
**Presentation:** 2
**Contribution:** 3
**Rating:** 4
**Confidence:** 3

**Summary:**

The paper presents GneissWeb, a 10T-token web dataset and an associated recipe for stage-1 LLM pretraining. Starting from FineWeb-V1.1.0, the authors add (1) sharded exact substring deduplication, (2) an ensemble of quality filters that includes two fastText classifiers, a category-aware readability filter (McAlpine-EFLAW), and a category-aware extreme-tokenized-documents filter (based on tokens/char and tokens/byte), and (3) lightweight category classifiers to apply category-specific thresholds. Across 1.4B/3B/7B Llama-style ablation models trained on 350B tokens, GneissWeb yields consistent gains over FineWeb and FineWeb-Edu-Score-2. It also reports 27% lower FLOPs to reach a 64% high-signal score at 7B. Stage-2 continuation from GneissWeb retains an advantage, and a Bloom-filter (~28 GB) is proposed to reproduce an approximation of the selected subset.

**Strengths:**

- The experimental results show clear improvements at multiple scales. Consistent gains at 1.4B/3B/7B on both high-signal and extended suites.
- This paper introduces a practical and scalable pipeline. Sharded exact substring dedup and ensemble filtering implemented with an open Data Prep Kit. The Bloom-filter reproduction path is pragmatic for large-scale users.
- Reporting FLOP reductions to a fixed quality target can be helpful.

**Weaknesses:**

- While each component is tested separately, the interactions among filters (and the precise contribution of threshold tuning) are not clearly separated. There is also a risk of overfitting the filtering thresholds to the evaluation sets.
- It seems that the pipeline leans on readability heuristics and tokenization extremes. There is limited analysis of the false-positive and false-negative rates and their downstream semantic impact (e.g., removal of niche yet valuable domains).
- Many of the data filtering methods in this paper are based on existing approaches, but the authors do not clearly explain their specific improvements or the details of their ensemble methods. This makes the paper’s contribution less clear.

**Questions:**

Please refer to the Weaknesses

---

> ### Author Response · Authors · 2025-11-22
> **Filter Interactions, Risk of Overfitting, and Novelty (Rebuttal 1/2)**
>
> We thank the reviewer for recognizing the practicality and scalability of our pipeline and the consistent improvements across multiple scales on both high-signal and extended evaluation suites. We appreciate their acknowledgement of our implementation using the open Data Prep Kit and the pragmatic Bloom-filter reproduction for large-scale users. We address their comments below.
>
> 1. **Interaction among Filters**
> - We conducted **ablation experiments on five different ensemble filtering strategies** to study how filters interact and contribute collectively (Sec. 2.2.6, App. D). These ensembles vary in logical rules combining: fastText classifiers combination, category-aware readability score filter, and category-aware extreme-tokenized documents filter.
> - Ensemble variants tested:
> | **Ensemble** | **Logical Rule** | **Description** |
> |--------------|-------------------|-----------------|
> | **Ensemble 1** | fastText AND readability AND extreme-tokenized | All filters must agree (strict AND). |
> | **Ensemble 2** | Any two of {fastText, readability, extreme-tokenized} | Retain if any two filters agree. |
> | **Ensemble 3** | fastText OR (readability AND extreme-tokenized) | Retain if fastText agrees OR both readability and extreme-tokenized agree. |
> | **Ensemble 4** | (fastText AND readability) OR (fastText AND extreme-tokenized), where fastText = AND of both classifiers | Same as GneissWeb rule but requires both fastText classifiers to agree. |
> | **Ensemble 5 (GneissWeb)** | (fastText AND readability) OR (fastText AND extreme-tokenized) | Retain if fastText agrees with either readability OR extreme-tokenized filter. |
> - The GneissWeb ensemble rule consistently outperforms individual filters and simpler combinations, demonstrating that filter interactions have a cumulative impact on quality.
> - We will add additional details on filter interactions and ensemble logic in the revised manuscript.
>
> 2. **Risk of overfitting the filtering thresholds to the evaluation sets**
> - We thank the reviewer for raising this important point. For threshold tuning,
>      - We use only 8 high-signal tasks (0-shot) out of 18 high-signal variants (Sec. 2.1).
>      - The remaining  the remaining high-signal tasks and the extended evaluation suite (covering diverse tasks) are used to asses generalization.
> - Tuning thresholds on a representative subset of benchmarks is a common practice in dataset-focused literature (e.g., FineWeb, Dolma, RefinedWeb, DCLM-Baseline).
> - Our extended evaluation results confirm that gains observed in ablations generalize across broader tasks, reducing the risk of overfitting.
> - We will explicitly highlight the use of 8 high-signal benchmarks for threshold tuning  in the revised manuscript, discuss the potential risk of overfitting, and clarify the role of the extended evaluation suite in validating generalization.
>
> 3. **Novelty of Data Filtering Methods**
> - We clarify that our work introduces **several new components** beyond existing approaches:
>     - *Readability Score Quality Filter (Novel)*: Utilizes linguistic clarity and human readability across domains to identify and exclude low-quality documents. This approach is not present in prior dataset recipes.
>     - *Extreme-Tokenized Document Removal (Novel)*: Leverages signals from both pre-tokenization and post-tokenization stages to detect and remove pathological tokenization cases. Not present in prior dataset recipes.
>    - *Ensemble Filtering Framework (Novel)*: Unlike prior works that rely on single model-based filters, we design an ensemble of heterogeneous filters combining fastText classifiers with our novel rule-based filters. Ensemble of quality filters enables us to achieve a fine-grained trade-off between quality and quantity, which is critical for Stage-1 scalability.
>     - *Domain-Aware Filtering (Novel)*: Incorporates document category information to avoid discarding high-quality domain-specific data, reducing bias introduced by uniform filtering.
> - *Use of Existing Methods*: We do use the fastText classifier from DCLM, but *extend it by training a complementary classifier* and integrating both into our ensemble alongside novel filters. This combination is **unique to the GneissWeb recipe** and improves performance compared to using fastText alone.
> - We will make these distinctions explicit in the manuscript, clarifying how our contributions differ from existing filtering methods and why the ensemble design is novel.

---

> ### Author Response · Authors · 2025-11-22
> **Analysis of Filters (Rebuttal 2/2)**
>
> 4. **Analysis of Readability and Extreme-Tokenized Filters**
>
> - We analyzed readability score and extreme-tokenized heuristics distributions across document categories (Appendix C.3 and C.4)
> - Observations:
>      - Distributions of certain categories differ from the overall distribution across categories. These specific categories tend to contain many documents with educational-style content, resulting in different scores than other categories.
>      - If a single global threshold is selected based on the overall score distribution and the same threshold is applied to all documents, there is a risk of losing high-quality documents (*false-negatives*).
>     - If the threshold is made lower to avoid potential loss of high-quality documents, then there is a risk of retaining lower quality documents (*false-positives*).
> - To mitigate this issue, we introduced *category-aware* thresholds. Category-aware thresholds help us preserve niche but valuable documents.
> - Ablation Results: we provide below results of ablations comparing single-threshold filtering versus category-aware filtering.
>
> | Filter Type                                                                                                  | High Signal Eval Score |
> |-------------------------------------------------------------------------------------------------------------|-------------------------|
> | Readability score quality filter: single threshold for all categories                                      | 52.35                  |
> | Category-aware readability score filter: lenient threshold for 4 key categories (“science”, “education”, “technology and computing”, and “medical health”) | 53.20                  |
> | Category-aware readability score filter: lenient threshold for 7 categories (“science”, “education”, “technology and computing”, “medical health”, “news and politics”, “business and finance”, and “personal finance”) | 52.67                  |
>
> - Key Findings:
>      - Category-aware filtering improves performance compared to a global threshold.
>      - Expanding lenient thresholds beyond the 4 key categories (e.g., adding “news and politics”, “business and finance”) did not yield further gains, suggesting that our approach effectively balances quality retention and semantic coverage.
>      - Observations for extreme-tokenized heuristics are similar: category-aware filtering improved performance compared to a single global threshold, and adding more categories to 4 key categories did not yield further gains.
>
> - We will highlight this analysis in the manuscript, discuss the risk of false positives/negatives, and clarify how category-aware filtering reduces the likelihood of discarding niche but valuable documents.

---

> > ### Comment · Reviewer_c8pE · 2025-11-27
> >
> > Thank you for your response. The revisions have addressed most of my concerns. I think the revised paper is now clearer in terms of the completeness of the analysis and the presentation of its contributions.
> >
> > Although most of the novelty lies in the application and combination of existing/simple metrics, the experimental results are convincing and demonstrate the effectiveness of the approach. Therefore, I raise my rating to 6.

---

### Official Review · Reviewer_vahU · 2025-11-05

**Soundness:** 4
**Presentation:** 3
**Contribution:** 3
**Rating:** 6
**Confidence:** 3

**Summary:**

The work introduces a 10T token LLM pretraining dataset called GneissWeb. In addition to traditional filters and deduplication, their data curation recipe includes new quality filters based on readability, tokenization, and category (domain). Using this dataset, the authors train models at 1.5B, 3B, and 7B scales, and show improvement in average performance across common benchmarks when compared to other contemporary open datasets.

**Strengths:**

The paper claims novelty in terms of several new filtering metrics being applied. These metrics are supported in the context of their training setup with ablations comparing performance against reasonable baselines. The authors are transparent about their methods and clarify all important details about their setup for data curation and training. The paper also touches on fairness/bias, training efficiency, and downstream LLM performance.

**Weaknesses:**

The ablations and comparisons against FineWeb appear thorough and convincing as a case for how the new specific filtering approach improves on the FineWeb recipe. However, the paper leaves unclear where this stands overall as a contribution to pretraining dataset construction.

One significant claim from the paper is that other competing filtering methods (like DC-LM Baseline) are more aggressive in quality filtering, and thus GneissWeb has the advantage of creating a larger pretraining dataset. However, this claim does not appear to be backed up by indicators of how GneissWeb would compare to these other methods given an equivalent starting dataset.

The paper also introduces new rule-based filters that show improvements in data quality. The effect of these filters seems underexplored, e.g., how do their tunable thresholds scale? How sensitive is performance to the thresholds of these parameters?

**Questions:**

1. How does GneissWeb compare to other recent open models trained at a similar scale? The focus of the baselines is FineWeb; are there other similar open datasets that should be compared against?
2. The paper seems to focus on cutting as little as possible during data curation, pointing out more aggressive filtering methods as a negative. However, the training experiments are all on a small subset of the dataset. Is there evidence that the performance would continue to scale at the same rate to the size of the full dataset?
3. On a related note, do you find any evidence of performance gains from making the filters more restrictive (i.e. changing the thresholds to filter more data)? Do the proposed filters continue to work well with stronger thresholds?

---

> ### Author Response · Authors · 2025-11-22
> **Positioning and Comparisons (Rebuttal 1/2)**
>
> We thank the reviewer for recognizing the novelty of our filtering metrics, the rigor of our ablation studies, and the clarity and transparency of our data curation and training methodology. We appreciate their acknowledgement of our efforts to consider fairness, training efficiency, and downstream performance. Below, we address the reviewer’s comments in detail.
>
> 1. **Contribution to Pre-Training Dataset Construction**
> - Our work tackles a fundamental challenge in LLM training: **constructing Stage-1 datasets that balance scale and quality for long-horizon pre-training**. While comparisons with FineWeb demonstrate clear improvements, the scope of our contribution is broader.
> - **Positioning within the Landscape**
>     - Most recent open-source datasets are *under 5T tokens* (e.g., Dolma, RefinedWeb, DCLM-Baseline, C4, The Pile, RedPajama, SlimPajama).
>     - GneissWeb provides ~10T tokens, filling a critical gap between smaller datasets and the largest ones (FineWeb at ~15T, RedPajamaV2 at ~30T). This makes GneissWeb a competitive Stage-1 solution for large-scale LLMs.
> - **Novel Methodology Beyond Existing Recipes**
>     - Beyond dataset size, our key contribution lies in the recipe itself, which introduces generalizable filtering strategies that go beyond simple model-based quality filtering.
>     - We propose two new quality filters: Readability Score Filter (targets linguistic clarity and coherence) and Extreme-Tokenized Documents Removal (identifies and removes pathological tokenization cases). We also employ domain-aware filtering by using document category information to preserve high-quality domain-specific data.
>     - These techniques enable fine-grained control over quality vs. quantity trade-offs and can enhance other dataset curation pipelines.
> - We will make this positioning explicit in the revised manuscript by clarifying that GneissWeb is not merely an improvement over FineWeb, but a **scalable, reusable recipe for Stage-1 dataset construction** with novel filtering strategies.
>
> 2. **Comparison with Other Datasets**
> - We primarily compared against FineWeb6 (15T) and FineWeb-Edu-Score-2 (5.4T), because these are widely regarded as state-of-the-art open datasets for Stage-1 training. They meet the *5T+ token threshold* and have demonstrated superior performance over several alternatives (App B).
> - To strengthen our case, we conducted **additional experiments** to compare GneissWeb against RedPajamaV2, TxT360, and Dolma. We follow our experimental setup (Sec 4) to train 1.4B parameter models on three random subsets of 350B tokens from these datasets, and evaluate on high-signal and extended benchmarks.
> - Models trained on GneissWeb outperform the models trained on other datasets across both evaluation suites:
>
> | **Dataset**  | **Tokens** | **High-Signal Eval** | **Extended Eval** |
> |---------|----|----|----|
> | Redpajama-v2 | 30T   | 57.70 $\pm$ 0.1 | 48.00 $\pm$ 0.34 |
> | TxT-360  | 4.8T   | 55.20 $\pm$ 0.2| 46.67 $\pm$ 0.31 |
> | Dolma  | 3T    | 54.18 $\pm$ 0.65| 47.24 $\pm$ 0.76 |
> | **GneissWeb** | 9.8T  | **58.40** $\mathbf{\pm}$ **0.19** | **48.82** $\mathbf{\pm}$ **0.27** |
>
> - We also examined Stage-2 pre-training using smaller, higher-quality datasets, DCLM-Baseline and FineWeb-Edu (Figure 4). Our results show that the GneissWeb model continues to demonstrate steeper scaling laws than the alternatives with consistently higher evaluation score, indicating that *Stage-1 gains persist into Stage-2*.
> - We will include the additional results to make it explicit that GneissWeb is competitive not only against FineWeb and FineWeb-Edu-Score-2, but also against RedPajamaV2, TxT360, and Dolma.
>
> 3. **Comparison with Other Models**
> - Similar to other dataset papers (e.g., FineWeb, Dolma, RedPajamaV2, RefinedWeb), our focus is to design a scalable data curation recipe. We specifically focus on a recipe for Stage-1 pre-training (~10T tokens). We do *not* aim to develop state-of-the-art models, requiring additional steps such as multi-stage training and long-context extension.
> - For context, we compare GneissWeb-trained models against **three open models of similar size (~1B parameters)**, each trained primarily on a single dataset with minimal additional steps (Figure 1b). This provides a fair comparison focused on Stage-1 pre-training quality.
>
> | **Model** | **Primary Dataset** | **Training Tokens** | **High-Signal Eval** | **Extended Eval** |
> |----|-----|-----|---|-----|
> | Pythia-1.4B  | The Pile | 300B | 54.12  | 45.96 |
> | Cosmo-1.8B | Cosmopedia | 180B    | 52.16  | 44.22 |
> | TinyLlama-1.1B | SlimPajama | 1.5T  | 57.58 | 48.36  |
> | **Gneiss-1.4B** | **GneissWeb** | **350B** | **58.67** | **49.20** |
>
> - We will explicitly highlight these model-level comparisons in the revised manuscript (Figure 1b) to clarify that GneissWeb is competitive both at the dataset level and when evaluated through open models trained at similar scale (focusing on Stage-1).

---

> ### Author Response · Authors · 2025-11-22
> **Experiment Scale, Comparisons, and Filter Sensitivity (Rebuttal 2/2)**
>
> 4. **Experiments on 350B Tokens**
>
> - Our experimental setup follows the standard practice in influential dataset-focused papers, and **our experimental scale (100B–350B tokens) aligns with several dataset papers (Dolma, RefinedWeb, FineWeb)**. For instance, FineWeb experiments are 350B tokens, Dolma experiments are 150B tokens, and RefineWeb experiments are 350B tokens.
> - Our results (Figures 3, 10, 11) show steeper scaling laws for models trained on GneissWeb, with GneissWeb models consistently outperforming alternatives throughout the training. Superiority of GneissWeb in ablation experiments at the Chinchilla-optimal 35B tokens translates well to 350B scale, indicating the robustness of our recipe. Prior work DCLM explicitly studied correlation between performance at smaller scales and larger scales, reporting high rank correlation.
> - We train on three random subsets of 350B tokens, effectively utilizing ~1T tokens from the datasets.  Unfortunately, training beyond 350B tokens is prohibitively expensive in terms of compute cost and time, which we will acknowledge in the revised manuscript.
>
> 5. **Comparison with Aggressive Filtering Approaches**
> - Context:
>      - Leading LLMs (e.g., .g., Llama-3: 15T, Gemma-2: 13T, SmolLM3: 11T) rely on Stage-1 datasets spanning tens of trillions of tokens for broad coverage, followed by later stages with higher-quality data.
>      - Our primary goal is to design a **Stage-1 recipe** that balances **scale and quality**, enabling long-horizon pre-training.
> - Aggressive Filtering Trade-off:
>      - Common Crawl remains the dominant source for Stage-1 data. Most datasets including (FineWeb, DCLM-Baseline, FineWeb-Edu) originate from Common Crawl.
>      - Datasets like DCLM-Baseline and FineWeb-Edu apply *highly aggressive filtering*, which improves quality but drastically reduces size, making them less suitable for Stage-1.
>      - GneissWeb starts from FineWeb (which itself originates from Common Crawl, similar to DCLM-Baseline) and applies a **novel ensemble of filters** to achieve a **better quality vs. quantity trade-off**.
> - *Ideal Comparison*:
>      - A perfect comparison would involve **full-scale pre-training (~10–12T tokens)** using GneissWeb versus alternatives (FineWeb, RedPajamaV2, or upsampling smaller high-quality datasets like DCLM-Baseline).
>      -  Unfortunately, experiments of this scale are not feasible within our compute budget. We will explicitly acknowledge this limitation in the revised manuscript.
> - Planned Revision:
>      - We will clarify that our approach is designed for *Stage-1 scalability*, where aggressive filtering methods cannot meet token requirements.
>      - We will emphasize that our experiments follow common practice for dataset-focused evaluations (typically 100B–350B tokens).
>
> 6. **Thresholds for new filters**
> - Readability Score Filter
>     - We performed ablations on different readability thresholds to identify the configuration that maximizes performance.
>     - Models were trained on 35B random tokens filtered using varying thresholds for four key categories and other categories.
>     - Evaluation was conducted on high-signal tasks (same setup as Sec. 2).
>     - Results show that thresholds of 70 (key categories) and 30 (other categories) achieve the best performance gain.
>
> | **Threshold for key categories** | **Threshold for other categories** | High-Signal Eval Score |
> |--|--|--|
> | **70** | **30** | **53.20** |
> | 70 | 17 | 52.89 |
> | 100 | 25 | 52.67 |
> | 70 | 40 | 52.35 |
>
> - Extreme-Tokenized Document Filter
>     - We also ablated thresholds for this filter to determine optimal bounds.
>     - Models were trained on 35B random tokens with different lower and upper bounds for key and other categories.
>     - Best performance was achieved with bounds (0.1, 0.5) and (0.22, 0.28) for key and other categories, respectively.
>
> | **Lower bound (key categories / other categories)** | **Upper bound (key categories / other categories)** | **High-Signal Eval Score** |
> |--|--|--|
> |  **0.1 / 0.22** | **0.5 / 0.28** | **52.85** |
> |  0.15 / 0.2 | 0.45 / 0.3 | 52.79 |
> |  0.17 / 0.2 | 0.4 / 0.32 | 52.31 |
>
> - We will include these ablation details in the revised manuscript to clarify sensitivity and justify parameter selection.

---

### Author Response · Authors · 2025-12-03
**Summary of Responses**

We thank the reviewers for their constructive feedback and recognition of the novelty, rigor, and scalability of our work. **We have uploaded the revised paper that incorporates the reviewers' comments with edits highlighted in blue.**

Below is a summary of our responses to help AC:

1. **Positioning and Contribution**
- We clarified that our work tackles a fundamental challenge in LLM training of constructing Stage-1 datasets that balance scale and quality for long-horizon pre-training.

- GneissWeb recipe a **scalable, reusable Stage‑1 dataset construction recipe** introducing novel filtering strategies. GneissWeb dataset provides ~10T tokens, filling a critical gap between smaller datasets (< 5T tokens, e.g., Dolma, RefinedWeb, FineWeb-Edu, DCLM-Baseline) and the largest ones (FineWeb at ~15T and RedPajamaV2 at ~30T).

- Our key contribution lies in the recipe itself, which introduces generalizable filtering strategies that go beyond simple model-based quality filtering.

     - We proposed **two new quality filters**, *Readability Score Filter* and *Extreme‑Tokenized Document Removal*, employ *Domain‑aware filtering* to preserve high‑quality niche content, and propose an *ensemble filtering framework* combining fastText classifiers with rule‑based filters for fine‑grained quality–quantity trade‑offs.

2. **Comparisons with Other Datasets and Models**
- We expanded experiments to include **RedPajamaV2, TxT360, and Dolma**, showing GneissWeb consistently outperforms alternatives across high‑signal and extended benchmarks. We elaborated on model‑level comparisons with open models of similar size (from Figure 1b) to confirm GneissWeb’s competitiveness for Stage‑1 pre‑training.

3.  **Experimental Scale and Limitations**
- Our **experiments (100B–350B tokens) align with standard practice** in dataset papers (e.g., Dolma, FineWeb, RefinedWeb). We acknowledged compute constraints preventing full 10T‑scale comparisons and clarified this in the manuscript.

4. **Filter Design and Threshold Justification**
- We provided **details on the ablation studies** for readability and extreme‑tokenized filters, demonstrating optimal thresholds and the benefits of category‑aware filtering to reduce false positives/negatives. Ensemble logic analysis shows cumulative gains over individual filters.

5. **Risk of Overfitting and Generalization**
- We clarified that our **threshold tuning was performed on a small subset** (8 out of 18 high‑signal tasks) to avoid bias, while the remaining high‑signal tasks and the extended evaluation suite were reserved for assessing generalization. This approach follows common practice in dataset‑focused literature. Extended evaluation confirmed that gains observed during ablations generalize across diverse tasks, mitigating overfitting concerns.

6. **Novelty and Practicality**

- We highlighted the originality of our filtering methods and ensemble design compared to prior works, and detailed implementation aspects (e.g., Data Prep Kit integration, Bloom‑filter reproduction) for scalability.

7. **Additional Clarifications**

- Included compute cost estimates for dataset creation and ablation experiments.

- Explained performance trends for math/code tasks as consistent with Stage‑1 objectives.

- Acknowledged limitations in starting from Common Crawl and the rationale for our approach.

We believe the rebuttal has strengthened our contributions and we hope to have addressed reviewers' questions.

---

### Meta-Review · Area_Chair_EiJA · 2026-01-03

**Summary:**

Overall, reviewers agreed that GneissWeb presents a strong, practical Stage-1 dataset construction recipe with convincing empirical gains, but raised concerns mainly around positioning, comparisons, and clarity of novelty, which were largely addressed in the rebuttal.

Reviewers initially questioned whether the work was more than an incremental improvement over FineWeb and how it fits into the broader landscape of pre-training datasets. The rebuttal clarified that the main contribution is a scalable, reusable Stage-1 recipe (not just a dataset), filling the gap between <5T and ultra-large (>15T) corpora, and emphasized the generality of the proposed filtering framework.

A key concern was limited comparison beyond FineWeb and the use of 100B–350B token training instead of full-scale (~10T) pretraining. The authors addressed this by adding comparisons with RedPajamaV2, TxT360, and Dolma, and by justifying the experimental scale as standard practice for dataset papers, while explicitly acknowledging compute limitations.

Regarding the novelty, some reviewers felt that components relied on existing heuristics (e.g., fastText). The authors clarified the novel aspects, e.g., the readability filter, extreme-tokenization filter, domain-aware thresholds, and the heterogeneous ensemble design, and explained how these go beyond prior recipes.

After rebuttal, reviewers converged toward borderline-positive to positive assessments, with one reviewer explicitly raising their score after revisions. The remaining concerns are acknowledged limitations rather than fatal flaws. Given the consistent empirical improvements, practical relevance, scalability, and strengthened clarity, the paper meets the bar for acceptance.

**Reviewer Concerns:**

Reviewer vahU: Concerns about the paper’s positioning, comparisons beyond FineWeb, and filter threshold sensitivity were addressed through clearer Stage-1 framing, added comparisons with RedPajamaV2/TxT360/Dolma, and detailed ablations, with the only remaining limitation being the lack of full ~10T-scale training due to compute constraints.

Reviewer c8pE: Questions regarding filter interactions, overfitting risk from threshold tuning, and methodological clarity were largely resolved via explicit ensemble logic analysis, restricted tuning protocols, and extended evaluations, leaving only a minor reservation that novelty lies more in the combination of methods than entirely new primitives.

Reviewer EiPH: Minor concerns about compute cost transparency, data source choice, and math/code performance were directly addressed in the rebuttal, with no substantive issues remaining outstanding.

**Reviewer Scores:**

Reviewer c8pE: 4 --> 6

Others may not be changed.

---

### Decision · Program_Chairs · 2026-01-26

Accept (Poster)